# Iron-sulfur cluster loss in mitochondrial CISD1 mediates PINK1 loss-of-function phenotypes

Sara Bitar[1], Timo Baumann[1], Christopher Weber[1], Majd Abusaada[1], Liliana Rojas-Charry[1], Patrick Ziegler[2], Thomas Schettgen[2], Isabella Eva Randerath[2], Vivek Venkataramani[3], Bernhard Michalke[4], Eva-Maria Hanschmann[5], Giuseppe Arena[6], Rejko Krueger[6,7,8], Li Zhang[1]*, Axel Methner[1]*

[1]University Medical Center of the Johannes Gutenberg-University Mainz, Institute for Molecular Medicine, Mainz, Germany; [2]Institute for Occupational, Social and Environmental Medicine, RWTH Aachen University, Aachen, Germany; [3]Comprehensive Cancer Center Mainfranken, University Hospital Würzburg, Würzburg, Germany; [4]Research Unit Analytical BioGeoChemistry, Helmholtz Zentrum München-German, Research Center for Environmental Health GmbH, Neuherberg, Germany; [5]Experimental and Translational Research, Department of Otorhinolaryngology, University Hospital Essen, Essen, Germany; [6]University of Luxembourg, Luxembourg Centre for Systems Biomedicine, Esch-sur-Alzette, Luxembourg; [7]Luxembourg Institute of Health (LIH), Strassen, Luxembourg; [8]Centre Hospitalier de Luxembourg (CHL), Luxembourg, Luxembourg

*For correspondence:
lizhang2017deu@gmail.com (LZ);
axel.methner@gmail.com (AM)

Competing interest: The authors declare that no competing interests exist.

**Abstract** Parkinson's disease (PD) is characterized by the progressive loss of dopaminergic neurons in the substantia nigra of the midbrain. Familial cases of PD are often caused by mutations of PTEN-induced kinase 1 (PINK1) and the ubiquitin ligase Parkin, both pivotal in maintaining mitochondrial quality control. CISD1, a homodimeric mitochondrial iron-sulfur-binding protein, is a major target of Parkin-mediated ubiquitination. We here discovered a heightened propensity of CISD1 to form dimers in Pink1 mutant flies and in dopaminergic neurons from PINK1 mutation patients. The dimer consists of two monomers that are covalently linked by a disulfide bridge. In this conformation CISD1 cannot coordinate the iron-sulfur cofactor. Overexpressing Cisd, the *Drosophila* ortholog of CISD1, and a mutant Cisd incapable of binding the iron-sulfur cluster in *Drosophila* reduced climbing ability and lifespan. This was more pronounced with mutant Cisd and aggravated in Pink1 mutant flies. Complete loss of Cisd, in contrast, rescued all detrimental effects of Pink1 mutation on climbing ability, wing posture, dopamine levels, lifespan, and mitochondrial ultrastructure. Our results suggest that Cisd, probably iron-depleted Cisd, operates downstream of Pink1 shedding light on PD pathophysiology and implicating CISD1 as a potential therapeutic target.

## Editor's evaluation

The study focuses on the role of CISD1 in Parkinson's disease (PD) and its relationship with the PINK1/Parkin pathway. The obtained data provide convincing evidence that apo-CISD1 may have a toxic function in Pink1 mutant flies and in dopaminergic neurons from patients with PINK1 mutation, thus serving as a critical mediator of Pink1-linked PD phenotypes. The findings provide valuable insights into the role of iron homeostasis and mitochondrial biology in PD.

**eLife digest** Parkinson's disease affects millions of people worldwide, causing progressively worse symptoms like stiffness, tremors and difficulty moving. These issues result from the death of neurons in the brain that produce the neurotransmitter dopamine. While most cases have no known cause, 10 to 15 per cent are due to inherited gene mutations. This includes mutations in the genes that code for the proteins PINK1 and Parkin which are essential for maintaining healthy mitochondria, the powerhouse of the cell.

Mutations in this quality control system affect a protein called CISD1, which sits within the outer surface of the mitochondria. CISD1 contains a cluster of iron and sulfur ions, and is involved in regulating iron levels and mitochondrial energy production. However, its role in inherited cases of Parkinson's disease, particularly those related to mutations in PINK1 and Parkin, is poorly understood.

To understand the impact of CISD1, Bitar et al. studied genetically modified fruit flies and dopamine-producing neurons from Parkinson's patients with PINK1 mutations. This revealed that losing PINK1 activity led to higher levels of CISD1 proteins which lacked the iron-sulfur cluster due to a bond forming between two CISD1 molecules.

Reducing levels of the CISD1-equivalent protein in the flies helped to alleviate most of the symptoms caused by PINK1 and Parkin gene mutations, such as difficulties climbing and impaired wing posture. These findings suggest that iron-depleted CISD1 contributes to the symptoms associated with Parkinson's disease, underscoring its potential as a drug target.

Drugs that target CISD1 already exist, which could ease the way for further research. Recent studies have shown that cases of Parkinson's related to mutations in PINK-1 share features with some non-inherited instances of the disease, suggesting that this approach could potentially benefit many patients.

## Introduction

Parkinson's disease (PD) is the fastest growing neurodegenerative disorder clinically characterized by rigor, tremor, and bradykinesia. These motor symptoms stem from the progressive degeneration of dopaminergic neurons in the midbrain's substantia nigra pars compacta (reviewed in *Pickrell and Youle, 2015*). While most cases are sporadic, familial forms of PD are linked to mutations in genes that encode proteins involved in mitochondrial quality control like PTEN-induced kinase 1 (PINK1) and Parkin (PRKN), providing valuable insights into the pathophysiology of PD (*Shulman et al., 2011*). In healthy cells, PINK1 is continuously imported into mitochondria, where it is rapidly degraded (*Narendra et al., 2008*). However, in unfit mitochondria with a reduced mitochondrial membrane potential ($\Delta\phi_m$), PINK1 import stalls. When PINK1 accumulates on damaged mitochondria, it phosphorylates various proteins, including the E3 ubiquitin ligase PRKN. This phosphorylation activates PRKN, enabling it to attach ubiquitin to mitochondrial proteins. This ubiquitination acts as a signal for damaged mitochondria to be identified and engulfed by autophagosomes, initiating their degradation through mitophagy. This selective elimination of compromised mitochondria safeguards overall mitochondrial health and function (reviewed in *Youle and Narendra, 2011*). Much of this mechanistic understanding has been elucidated in the fruit fly *Drosophila melanogaster* where *Pink1* and *Prkn* loss of function results in profound changes of the mitochondrial ultrastructure causing mitochondrial dysfunction, impaired flight and climbing ability, and a shortened lifespan (*Clark et al., 2006*; *Park et al., 2006*).

In addition to disturbances in mitochondrial quality control, disruptions in iron metabolism contribute significantly to the pathogenesis of PD (*Ma et al., 2021*). PD patients exhibit elevated iron levels within the substantia nigra pars compacta, and this iron accumulation correlates with the severity of the disease (*Dexter et al., 1987*; *Hirsch et al., 1991*). Notably, this iron dysregulation is apparent even in asymptomatic carriers of *PRKN* mutations (*Pyatigorskaya et al., 2015*), suggesting that it is an early event and correlates with mitochondrial dysfunction. Furthermore, human dopaminergic neurons lacking PINK1 display inhibition of the iron-dependent mitochondrial aconitase, leading to accumulation of NAD$^+$ and depletion of essential metabolites (*Bus et al., 2020*). In flies, *Pink1* deficiency similarly results in inactivation of mitochondrial aconitase by impairing its labile iron-sulfur cluster, leading to oxidative distress and mitochondrial swelling (*Esposito et al., 2013*; *Wan*

*et al., 2020*). These findings highlight the relevance of iron dysregulation in the PD development and progression.

An important target protein of PINK1/PRKN crucial for iron homeostasis and redox homeostasis is CISD1, also known as mitoNEET (reviewed in *Mittler et al., 2019*). CISD1 is a homodimeric protein N-terminally inserted into the outer mitochondrial membrane facing the cytosol (*Wiley et al., 2007a*). CISD1 is ubiquitinated by PRKN (*Sarraf et al., 2013*) and its levels are reduced upon dissipation of $\Delta \psi_m$ (*Chan et al., 2011*; *Cunningham et al., 2015*; *Lazarou et al., 2013*; *McWilliams et al., 2018*; *Narendra et al., 2012*; *Okatsu et al., 2012*). Moreover, CISD1 forms a complex with PRKN, and this interaction is markedly increased by activation of PINK1 and ubiquitination (*Narendra et al., 2013*; *Sarraf et al., 2013*). In *Drosophila*, the homolog of CISD1 is the most strongly ubiquitinated protein in response to Prkn overexpression in vivo (*Martinez et al., 2017*).

Functionally, CISD1 exists as a homodimer, and each monomer of CISD1 harbors an 2Fe/2S (2 iron/2 sulfur) cluster coordinated by three cysteines (Cys72,74,83) and one histidine (His87). In contrast, most conventional 2Fe/2S clusters are coordinated by four cysteines or two cysteines and two histidines (*Wiley et al., 2007b*). This peculiarity renders the CISD1 2Fe/2S cluster more susceptible to oxidation. It was thus proposed that the transfer of the iron-sulfur cluster from CISD1 to stress-sensitive Fe/S proteins constitutes a cellular adaptive response that helps recovery from oxidative injury (reviewed in *Mittler et al., 2019*). In support of this hypothesis, it has been demonstrated that the oxidized 2Fe/2S cluster of CISD1 can be transferred to iron regulatory protein 1 (IRP1) (*Ferecatu et al., 2014*), a bifunctional protein that can switch between two distinct enzymatic activities based on the availability of iron. When iron levels are sufficient, IRP1 binds an 2Fe/2S cluster, converting it into a functional aconitase enzyme. When iron levels are low, IRP1 loses its 2Fe/2S cluster and undergoes a conformational change. In this state, known as the 'apo' form, IRP1 functions as an RNA-binding protein and binds to specific RNA sequences called iron-responsive elements (IREs) located in the untranslated regions of target mRNAs (reviewed by *Rouault and Maio, 2017*). CISD1 is capable of recycling the cytosolic apo-IRP1 to holo-aconitase by donating its 2Fe/2S cluster (*Camponeschi et al., 2017*). The fate of the remaining apo-CISD1 is unknown. In addition to its role in iron metabolism and defense against oxidative distress, CISD1 is also involved in mitochondrial bioenergetics and in the regulation of mitochondrial morphology. Cardiac myocytes from *Cisd1* knockout (KO) mice exhibit a reduced mitochondrial oxidative capacity (*Wiley et al., 2007a*), akin to KO mouse embryonic fibroblasts (MEFs) where this dysfunction is attributed to a reduction in the total cellular mitochondrial volume (*Vernay et al., 2017*).

In *Drosophila*, there is a single ortholog of CISD1 initially named Cisd2 because it is slightly more similar to the mammalian CISD2 than to CISD1 (*Jones et al., 2014*) protein. Human CISD2 is located at the membrane of the endoplasmic reticulum (ER) (*Wang et al., 2014*). Overexpression of Cisd2 (this publication used the alternative name Dosmit) in fly muscles is detrimental and results in mitochondrial enlargement and the formation of double-membraned vesicles containing cytosolic proteins within mitochondria (*Chen et al., 2020*), implicating Cisd2 in the regulation of mitochondrial morphology. Similar to human CISD1, fly Cisd2 is an Fe/S protein and all amino acids linked to cluster coordination are conserved. In addition, fly Cisd2 (this publication used the alternative name MitoNEET) interacts with IRP1 (*Huynh et al., 2019*) further linking Cisd2 to iron homeostasis. Together these findings suggest that fly Cisd2 rather resembles the human mitochondrial homolog CISD1 and to avoid confusion, we will henceforth use Cisd for the fly ortholog. Based on the involvement of mammalian CISD1 in mitochondrial bioenergetics and quality control, iron metabolism and defense against oxidative distress – all hallmarks of PD – we aimed to investigate CISD1's role in PD using both fly and mammalian model systems.

In our study, we identified CISD1, particularly its iron-depleted apo form, as a downstream mediator of the PD pathology induced by the loss of Pink1 and Prkn function. Apo-CISD1 accumulates in PINK1 patient-derived human dopaminergic neurons and apo-Cisd in mutant *Pink1* flies. Complete loss of *Cisd* rescues all detrimental effects of *Pink1* loss of function on climbing ability, wing posture, dopamine levels, lifespan, and mitochondrial ultrastructure. In *Prkn* mutant flies, *Cisd* deficiency ameliorates climbing ability and wing posture impairments, but does not mitigate the reduction in lifespan. These findings strongly suggest that the buildup of iron-depleted CISD1 may contribute to the pathophysiology of both familial and sporadic PD.

## Results

### Increased CISD1 dimer formation in human dopaminergic neurons from PD patients with a PINK1 Q456X mutation

To investigate a possible involvement of CISD1 and its homolog CISD2 in familial PD, we studied their expression levels in differentiated dopaminergic neurons from two distinct patients afflicted with autosomal-recessive PD caused by *PINK1* Q456X mutation. This specific mutation introduces a premature stop codon, leading to increased apoptosis of tyrosine hydroxylase-positive dopaminergic neurons, accompanied by increased levels of alpha-synuclein in the surviving dopaminergic neurons and astrogliosis (*Jarazo et al., 2022*). Both *PINK1* mutant and gene-corrected induced pluripotent stem cells (iPSCs) were first converted into neuroepithelial stem cells (NESCs) and then further differentiated into midbrain-specific dopaminergic neurons as previously described (*Reinhardt et al., 2013*). Notably, *PINK1* Q456X neurons expressed diminished *PINK1* mRNA levels (*Figure 1A*) and reduced levels of tyrosine hydroxylase (as depicted in exemplary blots from our differentiations in *Figure 1B*, with quantification of different independent differentiations reported in *Jarazo et al., 2022*).

We then employed immunoblotting to concurrently assess the abundance of both mitochondrial CISD1 and ER CISD2 proteins, because both proteins are recognized by our antibody (*Martinez et al., 2024*; *Wang et al., 2014*). The ability of this antibody to identify both homologs was previously demonstrated by subcellular fractionation (*Wang et al., 2014*) and specific siRNA-mediated knockdown of either *CISD1* or *CISD2* (*Martinez et al., 2024*). It is also noteworthy that both proteins form homodimers with high stringency even on reducing gels (*Antico et al., 2021*; *Paddock et al., 2007*). *Cisd1* KO and wildtype (WT) MEFs (*Vernay et al., 2017*) served to identify the correct bands (*Figure 1C/C'*). Our immunoblots revealed no changes in the abundance of CISD1 or CISD2. However, we clearly observed an increased presence of the CISD1 dimer in mutant *PINK1* neurons. Notably, both patients exhibited an elevated dimer/monomer ratio with variations in each individual differentiation in patient #1, and a more pronounced effect in patient #2, implying the presence of additional patient-specific factors. When pooled together, this analysis revealed a significant increase in CISD1 dimerization in mutant *PINK1* neurons compared to gene-corrected controls (*Figure 1D*). This heightened propensity for dimer formation therefore appears to be relevant for human PD.

### CISD1 lacking its iron-sulfur cluster has a higher propensity to dimerize

To next clarify under which conditions CISD1 exhibits a heightened propensity to run as a dimer, we developed a split luciferase assay to investigate CISD1 dimerization in living cells in a dynamic and reversible manner (*Dixon et al., 2016*). We fused CISD1 with the two small subunits of NanoLuc luciferase, enabling us to quantify dimerization by measuring luminescence. Given the labile nature of CISD1's iron-sulfur cluster due to its unconventional coordination by one histidine and three cysteines, we hypothesized that the presence or absence of this cluster might influence dimerization. We therefore mutated histidine 87 to cysteine (H87C) which results in a more stable 2Fe/2S complex coordination (*Conlan et al., 2011*). Conversely, the mutation of cysteine 83 to serine (C83S) completely abolishes the ability of CISD1 to bind the 2Fe/2S complex. By comparing the dimerization patterns of the CISD1 variants, we observed a significant increase in dimerization for C83S CISD1, the mutant incapable of binding the iron-sulfur cluster (*Figure 2A*). This finding suggested that iron-depleted CISD1 has a higher propensity for dimerization. To further validate these findings, we next employed an in silico approach to calculate the molecular hydrophobicity potential and the angle of rotation between two CISD1 WT models or two CISD1 C83S models (*Polyansky et al., 2014*). This analysis unveiled a larger surface-surface contact between the C83S monomers, indicating a closer binding compared to WT CISD1 (*Figure 2B*). This aligns with the results from the dimerization assay (*Figure 2A*).

Given that CISD1 is an important target of the PINK1/PRKN machinery, we hypothesized that conditions that induce mitophagy might yield a similar increase in dimerization. To investigate this, we conducted our dimerization assay in cells subjected to a 2 hr treatment with the uncoupling agent carbonyl cyanide-*p*-trifluoromethoxyphenylhydrazone (FCCP), a combination of the complex V inhibitor oligomycin and the complex III inhibitor antimycin A (A/O), or the iron chelator deferiprone, which triggers a distinct form of mitophagy independent of PINK1 and PRKN (*Allen et al., 2013*). The 2 hr treatment was chosen to exclude alterations caused by degradation of CISD1. Remarkably, our results demonstrated that under these conditions only iron chelation resulted in increased dimerization (*Figure 2C*) that mirrored the increased dimerization seen in the C83S mutant, which is unable

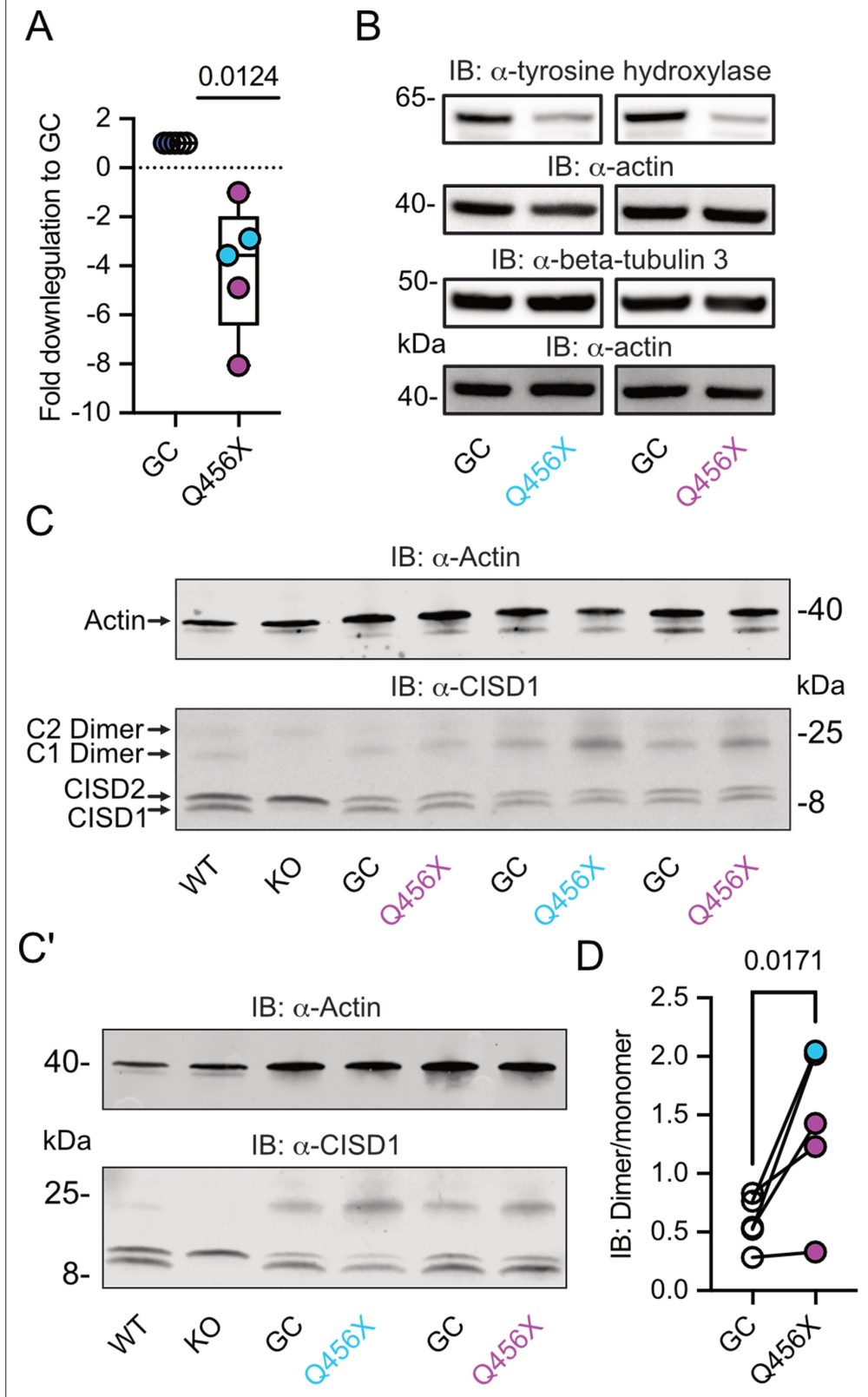

**Figure 1.** Increased CISD1 dimer formation in human dopaminergic neurons from Parkinson's disease (PD) patients with a *PINK1* Q456X mutation. (**A**) Quantitative PCR analysis of *PINK1* mRNA expression in induced pluripotent stem cell (iPSC)-derived dopaminergic neurons from PD patients carrying the *PINK1* Q456X mutation, colors indicate two different patients. *ACTB* served as housekeeping control. Expression levels were normalized to

*Figure 1 continued on next page*

*Figure 1 continued*

gene-corrected (GC) controls. (**B**) Immunoblots of tyrosine hydroxylase as a marker for dopaminergic neurons and beta-tubulin 3 as a general neuronal marker. Actin served as loading control, size is indicated. (**C, C'**) Immunoblots of CISD1 and CISD2 with lysates from different patients, GC controls, and independent differentiations over 28 days in vitro. Monomeric and dimeric forms of both proteins are indicated by arrows. CISD1 knockout (KO) and wildtype (WT) mouse embryonic fibroblasts served to identify the correct bands. Actin served as loading control, size is indicated. (**D**) Quantification of the CISD1 dimer/monomer ratio reveals an increased ratio in *PINK1* Q456X dopaminergic neurons over gene-corrected controls. Each data point corresponds to an independent differentiation over 28 days and the colors designate the two different patients. Data in A were normalized to the respective gene-corrected control and are presented as box and whisker plots with the box extending from the 25th to the 75th percentile and the whiskers from the minimum to the maximum value depicting all individual data points. Data points in D correspond to the mean of two independent technical replicates of five independent differentiations from two patients and are unnormalized. Statistical analysis was done using the one-sided t test in A and a nested t test in D, p values are indicated.

The online version of this article includes the following source data for figure 1:

**Source data 1.** Raw and labeled blots of iPSC-derived dopaminergic neurons from PD patients carrying the PINK1 Q456X with their gene corrected controls.

**Source data 2.** Raw data of PCR and immunoblot quantifications of *Figure 1*.

to coordinate the 2Fe/2S complex (*Figure 2A*). To corroborate this, we expanded our analysis to WT MEF cells treated with deferiprone. Immunoblotting revealed increased dimer formation and even the presence of a CISD1 multimer under these conditions (*Figure 2D*). Consequently, we conclude that the increased dimer formation in the *PINK1* mutant neurons likely represents the presence of apo-CISD1, a form of CISD1 devoid of its iron-sulfur cluster, and that this is caused by changes in the iron homeostasis.

## Increased Cisd dimer levels in *Pink1* mutant flies reflect an increased intermolecular disulfide bond between two Cisd molecules

We next investigated whether Pink1 loss of function in flies mirrors the human situation and impacts Cisd dimerization or possibly abundance. Pink1[B9] flies (*Park et al., 2006*) are on the w[1118] background and have a large deletion of the coding region and almost no remaining mRNA expression (*Figure 3—figure supplement 1*). At the mRNA level, we observed no changes in Cisd mRNA levels in Pink1[B9] flies (*Figure 3A*). We then immunoblotted samples from young (3-day-old) and old (8-week-old) WT w[1118] and Pink1[B9] flies and noted that fly Cisd, similar to mouse CISD1, runs as a monomer and dimer even on reducing gels containing dithiothreitol (DTT) (*Figure 3B*). We quantified total Cisd protein levels, i.e., monomer plus dimer and normalized them to actin. Additionally, we assessed the dimer/monomer ratio. This analysis unveiled reduced total Cisd protein levels in young Pink1 mutant flies compared to WT flies (*Figure 3C*). We were unable to replicate the previously reported age-dependent increase in Cisd abundance in WT flies (*Chen et al., 2020*) and Pink1[B9] (*Martinez et al., 2024*) flies although this was evident in many but not all samples indicating additional factors. When assessing the dimer/monomer ratio, we observed a significant increase in dimer levels in old Pink1[B9] flies compared to young Pink1[B9] flies and to old WT flies (*Figure 3D*). Age had no effect on dimerization of WT flies (*Figure 3D*). Thus, Pink1 mutant flies manifest analogous alterations to those observed in dopaminergic neurons from patients afflicted by familial PD due to PINK1 Q456X mutation (*Figure 1D*). To elucidate the nature of the dimer, we next used extremely reducing conditions of prolonged incubation with higher concentrations of DTT and tris-(2-carboxyethyl)-phosphine as a disulfide-bond breaking agent followed by boiling. This resulted in the loss of dimeric and an increase of monomeric Cisd (*Figure 3E*). We also added the thiol alkylating agent *N*-ethylmaleimide (NEM) during lysis and performed non-reducing immunoblotting in old Pink1[B9] flies compared to old WT flies. NEM blocks all reduced sulfhydryl (-SH) groups of cysteine residues and thereby prevents the formation of disulfide bonds due to non-specific oxidation during the lysis process (*Figure 3F*). We also observed the increased dimer formation in old Pink1[B9] flies under these conditions (*Figure 3F*). Together these experiments indicate that the increased dimerization in old Pink1[B9] flies is most probably caused by the formation of an intermolecular disulfide linking two Cisd molecules.

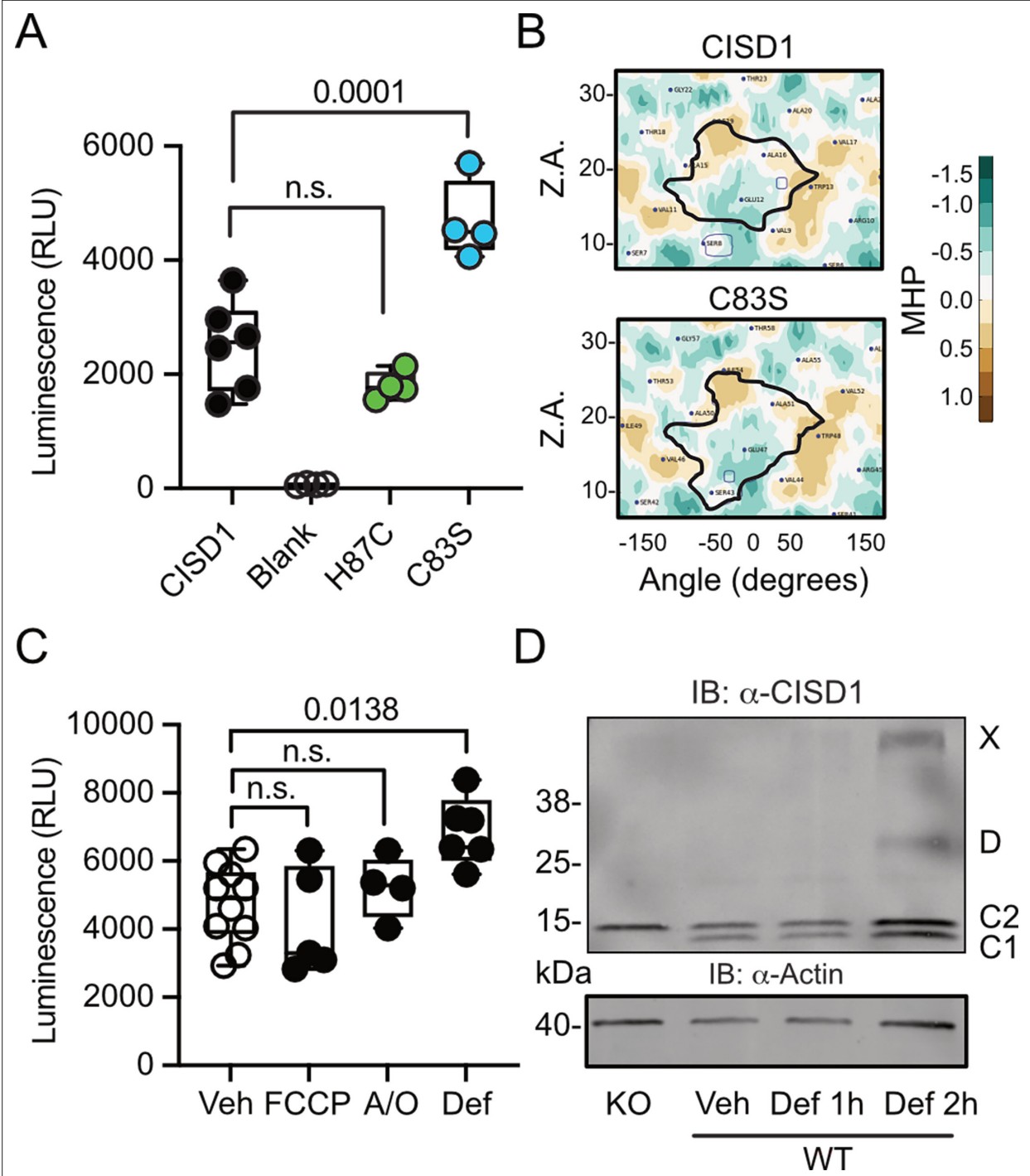

**Figure 2.** CISD1 lacking its iron-sulfur cluster has a higher propensity to dimerize. (**A**) Cells were transiently transfected with wildtype (WT) or point-mutated CISD1 tagged with split NanoLuc fragments. Luminescence was quantified after 48 hr. (**B**) 2D maps of the molecular hydrophobicity potential (MHP) on the peptide surfaces in two CISD1 models or two C83S models. Dimerization interfaces are outlined. Axis values correspond to the rotation angle around the helical axis (α) and the distance along the latter (Z), 2D maps are colored according to MHP arbitrary units. The surface-surface area is indicated by a solid line. (**C**) Cells were transiently transfected with WT CISD1 tagged with split NanoLuc fragments and treated for 2 hr with 1 μM carbonyl cyanide-*p*-trifluoromethoxyphenylhydrazone (FCCP), 2.5 μM antimycin and oligomycin (A/O), or 1 mM deferiprone (Def) before addition of substrate and quantification of luminescence. (**D**) Immunoblot of WT mouse embryonic fibroblast (MEF) cells treated for the indicated period of time with 1 mM Def or vehicle (Veh). Knockout (KO) cells served as control for antibody specificity and actin as loading control. Each data point in A and C is from five independent experiments done in triplicates. Data are presented as box and whisker plots with the box extending from the 25th to the 75th

*Figure 2 continued on next page*

*Figure 2 continued*

percentile and the whiskers from the minimum to the maximum value depicting all individual data points. Statistical analysis was done using one-way ANOVA, p values are indicated.

The online version of this article includes the following source data for figure 2:

**Source data 1.** Immunoblots of WT MEF cells treated with deferiprone or vehicle.

**Source data 2.** Raw data of the dimerization assays in *Figure 2*.

## Altered iron homeostasis reflecting a more pro-oxidative state in old *Pink1^{B9}* flies

Having demonstrated that iron dyshomeostasis induced by the iron-depleting agent deferiprone results in increased CISD1 dimer formation in fibroblasts, we next investigated iron dynamics in *Pink1* mutant flies compared to controls by employing mass spectrometry. This revealed a heightened total iron content in Pink1 mutants (*Figure 3G*). Delving deeper with capillary electrophoresis-inductively coupled plasma mass spectrometry (CE-ICP-MS), we then identified an enrichment of redox-active $Fe^{2+}$ ions in these mutants compared to the more subdued presence of redox-inactive $Fe^{3+}$ (*Figure 3H*). In the Fenton reaction, ferrous iron ($Fe^{2+}$) forms hydroxyl radicals (•OH), a highly reactive and destructive free radical, and ferric iron ($Fe^{3+}$) in the presence of hydrogen peroxide. *Pink1^{B9}* flies are therefore in a more pro-oxidative state. *Pink1^{B9}* flies were also dramatically more susceptible to iron depletion with deferiprone. In these experiments, we fed developing WT and Pink1 mutant flies food supplemented with 65 µM deferiprone, a concentration that did not induce widespread mitophagy in *Pink1^{B9}* flies (*Lee et al., 2018*), and quantified the number of adult flies eclosing from pupae. This revealed that iron depletion also showed a trend toward reduced eclosion of WT flies but nearly completely abrogated the eclosion of *Pink1* mutant flies (*Figure 3I*). Collectively, these results support a connection between iron dyshomeostasis, redox imbalance, and the presence of an intermolecular disulfide bond between two molecules of Cisd leading to apo-Cisd, although establishing causality in this relationship is challenging.

## Overexpression of Cisd and apo-Cisd in *Drosophila* is detrimental

To investigate potential detrimental effects of apo-Cisd and whether these would be more pronounced in the absence of functional Pink1, we next overexpressed either *Cisd* or a mutated *Cisd C/S* where we mutated all cysteine residues that coordinate the Fe/S cluster to serines (C100S, C102S, and C/S) in WT *w^{1118}* and *Pink1^{B9}* mutant flies using the mild ubiquitous Gal4 driver *daughterless* (*da*). *Cisd C/S* constitutes a constitutive apo-Cisd variant. Flies overexpressing mitochondrially targeted GFP (mtGFP) served as control. Ectopic expression of *Cisd* in muscle tissue was already shown previously to reduce lifespan by enlarging mitochondria and the formation of intramitochondrial vesicles (*Chen et al., 2020*). We chose negative geotaxis (the ability to climb) and lifespan – known hallmarks of the *Pink1^{B9}* phenotype (*Clark et al., 2006*) – as readouts and found that *Pink1^{B9}* flies have reduced negative geotaxis and lifespan as expected. Ubiquitous overexpression of Cisd and apo-Cisd aggravated these phenotypes in WT and Pink1 mutant flies (*Figure 4A*). *Cisd C/S* overexpression was, however, associated with a more profound reduction in climbing speed compared to WT Cisd, while the effects on lifespan were similarly detrimental (*Figure 4A*). Additionally, the vast majority of all *Cisd C/S*-overexpressing flies were not capable of inflating their wings (*Figure 4B*), a rather non-specific phenotype which can result from genetic mutations, environmental conditions, and developmental defects. This phenotype did not occur in WT *Cisd* or *mitoGFP*-overexpressing flies (*Figure 4B*). Based on the above-observed more pro-oxidative state in *Pink1^{B9}* flies (*Figure 3E/F*), we next quantified glutathione *S*-transferase S1 (*GstS1*) mRNA levels in flies overexpressing Cisd and apo-Cisd. *GstS1* is part of the antioxidant response and was identified previously in an unbiased genetic screen for genes that modify *Prkn* phenotypes (*Greene et al., 2005*). It was also shown that overexpression of *GstS1* in dopaminergic neurons suppresses neurodegeneration in *Prkn* mutants (*Whitworth et al., 2005*). Interestingly, newly eclosed flies expressing *C/S Cisd* displayed significantly higher levels of *GstS1* mRNA in *w^{1118}* in line with a compensatory increase in the defense against oxidative distress induced by apo-Cisd (*Figure 4C*). *Pink1^{B9}* flies, in contrast, were not capable of raising this antioxidant defense in response to apo-Cisd expression (*Figure 4C*) which might explain part of the additional toxic effect. In summary, our findings suggest an adverse impact of both WT and iron-depleted Cisd on climbing

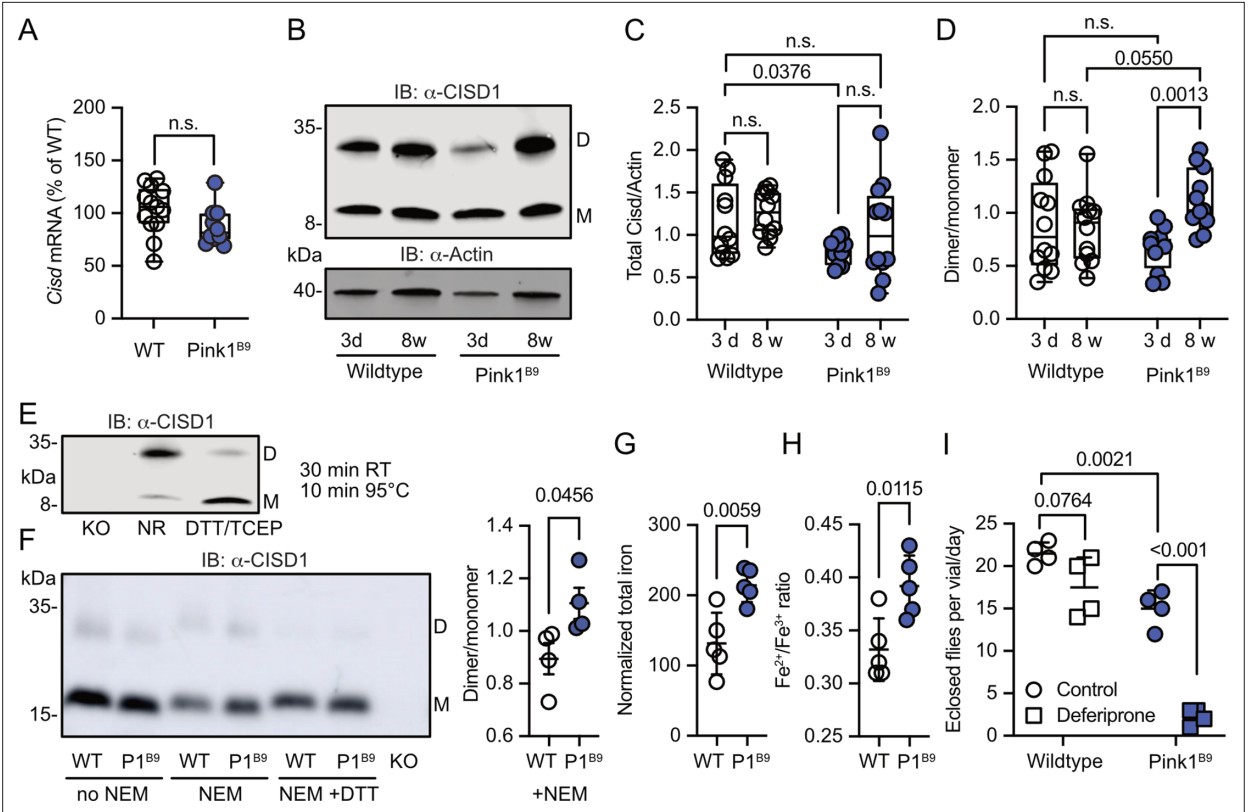

**Figure 3.** Increased Cisd dimer levels and altered redox and iron homeostasis in *Pink1* mutant flies. (**A**) Quantitative PCR analysis of *Cisd* mRNA levels in *Pink1$^{B9}$* flies. *RpL32/Rp49* served as housekeeping control. Each dot represents the mean relative transcriptional level of a sample of five flies. The values were normalized to the mean value of wildtype (WT) flies. (**B**) Immunoblot analysis of Cisd abundance in the indicated fly strains at different ages, 3 days (d) and 8 weeks (w). Actin served as loading control, size is indicated. (M) Cisd monomers and (D) dimers. (**C**) Total Cisd (D+M) normalized to actin. Each dot represents the mean of two technical replicates from n=12 independent samples of two male flies. (**D**) Dimer/monomer ratio of the same samples. (**E**) Immunoblot analysis of Cisd protein levels in w1118 flies. Samples were exposed to a non-reducing buffer (NR) or reducing buffer containing 100 mM dithiothreitol (DTT) and 50 mM TCEP. Cisd knockout (KO) lysates served as negative control for the Cisd bands. Actin served as loading control, size is indicated. (M) Cisd monomers and (D) dimers. (**F**) Cisd immunoblot analysis in the absence and presence of *N*-ethylmaleimide (NEM) and the reducing agent DTT and quantification of the dimer/monomer ratio in the presence of NEM. (**G**) Total iron content and (**H**) $Fe^{2+}/Fe^{3+}$ ratio measured via electrophoresis-inductively coupled plasma mass spectrometry (CE-ICP-MS) in WT and *Pink1$^{B9}$* flies. Each dot represents a group of three male flies. n=5. (**I**) Flies of the same age, number, and sex ratio were allowed to mate and lay eggs in normal food or food prepared with a 65 μM final concentration of deferiprone. Each dot represents the average number of eclosed flies per vial and day for 3 consecutive days. Data are presented in A/C/D as box and whisker plots with the box extending from the 25th to the 75th percentile and the whiskers from the minimum to the maximum value depicting all individual data points and as scatter plots with the mean and SD in F–I. Statistical analysis was done using one-way ANOVA in A, two-way ANOVA in C/D, and Student's t test in F–I, p values are indicated.

The online version of this article includes the following source data and figure supplement(s) for figure 3:

**Source data 1.** Raw and labeled blots of young and old wildtype and Pink1B9 flies blotted against CISD1 or actin as well as flies treated with or without NEM or DTT.

**Source data 2.** Raw data of PCR and immublotting and iron measurments quantification as well as flies eclosion from *Figure 3*.

**Figure supplement 1.** Validation of Pink1 knockout flies.

**Figure supplement 1—source data 1.** Raw and labeled gels from a gel electrophoresis validating Pink1 knockout flies.

**Figure supplement 1—source data 2.** Raw data of the quantification of the Pink1 knockout flies.

speed and lifespan. Under conditions of robust expression, apo-Cisd appears to exert an even more pronounced deleterious effect possibly by influencing redox homeostasis.

### *Cisd* depletion rescues *Pink1* mutant phenotypes

To clarify whether depleting the potentially detrimental (apo) Cisd could mitigate Pink1 pathology, we first used a ubiquitous knockdown of *Cisd* in *Pink1$^{B9}$* mutant flies by RNAi driven by the tubulin driver

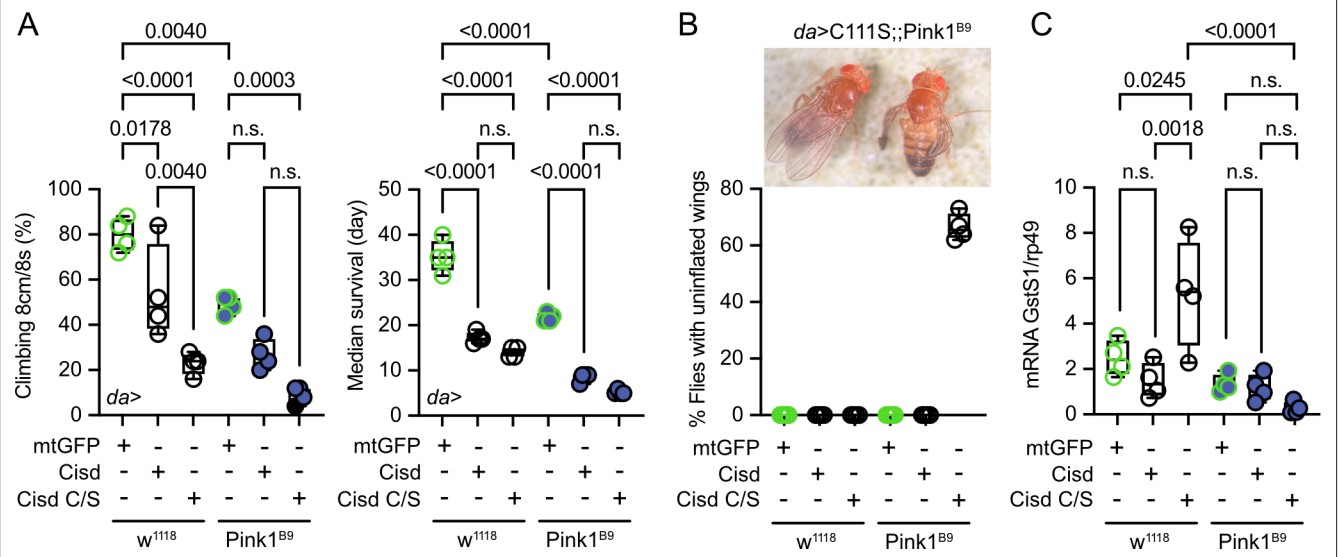

**Figure 4.** Overexpression of *Cisd* and *apo-Cisd* in *Drosophila* is detrimental. (**A**) Climbing assay and median survival of flies overexpressing mitochondrially targeted GFP (mtGFP), *Cisd,* or *Cisd C/S* (all Fe/S coordinating cysteines mutated to serines) on standard food. Overexpression was achieved using the UAS-Gal4 system and the mild ubiquitous *daughterless (Da)* driver line on a w[1118] or Pink[B9] background. In the climbing assay, each dot represents the percentage of a group of 25 flies that successfully climbed more than 8 cm within 8 s. For the median survival, the number of dead flies was scored every 2 days and fresh food was supplied. Each dot represents the median lifespan of a group of 25 flies. (**B**) Representative image and quantification of the % of eclosed flies with non-inflated wings overexpressing *Cisd C/S* on a Pink[B9] background 3 days after eclosion. (**C**) Quantitative PCR analysis of *GstS1* mRNA levels in newly eclosed flies. Rp49 served as housekeeping control. Each dot represents the mean relative transcriptional level of a sample of five flies. The values were normalized to the mean value of control flies (da-Gal4>*mtGFP*). Statistical analysis was done using one-way ANOVA, p values are indicated.

The online version of this article includes the following source data for figure 4:

**Source data 1.** Raw data of the quantification of climbing and lifespan and wing phenotype as well as quantitative PCR anaylsis of flies overexpressing Cisd or CS mutant on wildtype or Pink1B9 background.

**Source data 2.** Image of flies under the microscope to show the uninflated wing phenotype in the flies overexpressing the CS mutant on a Pink1B9 background.

focusing on lifespan as a readout. Because *Pink1* resides on the fly heterosome, hemizygous male flies represent a complete KO while heterozygous female flies still retain one functional allele and serve as controls. Knockdown of *Cisd* led to an approximately 60% reduction of Cisd protein levels (**Figure 5—figure supplement 1A, B**). Importantly, this knockdown effectively rescued the reduced lifespan (**Figure 5—figure supplement 1C**) of *Pink1* mutant males, suggesting a detrimental effect of *Cisd* in flies lacking *Pink1*. To rule out any potential unspecific effects resulting from RNAi, we subsequently generated a complete KO of the entire genomic sequence of *Cisd* in w[1118] flies using CRISPR/Cas9 technology (**Figure 5—figure supplement 2A**). Flies devoid of Cisd protein expression exhibited no discernible difference in climbing ability up to 30 days of age (**Figure 5—figure supplement 2B**) and only a minor reduction in lifespan (**Figure 5—figure supplement 2C**). This contrasts with a previously reported Cisd null mutant generated by transposon integration (*Cisd2[G6528]*), which was shown to increase lifespan (**Chen et al., 2020**). The complete KO allowed us to observe that Cisd not only runs as a monomer (M), a dimer (D) as shown and quantified earlier, but also as a weak multimer (X) (**Figure 5A**). Given the unavailability of functional antibodies against fly Pink1, we quantified *Pink1* mRNA levels to verify the loss of *Pink1* expression in *Pink1[B9]* flies and to investigate potential dysregulations in *Cisd* KO flies. *Pink1* levels were reduced in *Pink1[B9]* but unchanged in *Cisd* KO flies (**Figure 5B**). We then investigated the effect of complete *Cisd* depletion on the phenotypes described in mutant *Pink1[B9]* flies including reduced dopamine levels, lifespan, climbing ability, a morphological wing phenotype caused by atrophy of the wing muscles, and an altered mitochondria ultrastructure (**Clark et al., 2006**; **Park et al., 2006**). As expected, *Pink1[B9]* flies exhibited reduced climbing ability (**Figure 5C**), an abnormal wing posture (**Figure 5D**), a decreased lifespan (**Figure 5E**), and diminished dopamine levels (**Figure 5F**) in line with prior reports (**Clark et al., 2006**; **Park et al., 2006**). We then

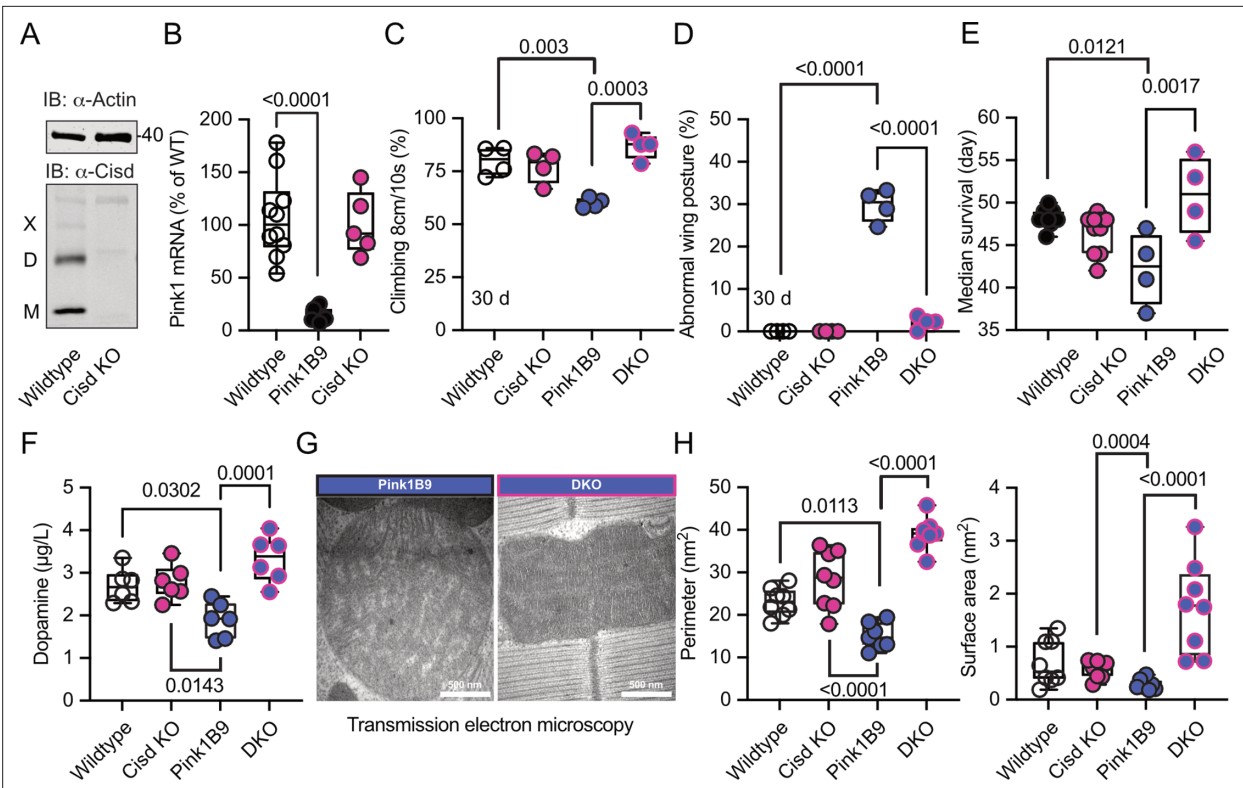

**Figure 5.** *Cisd* depletion rescues *Pink1* mutant phenotypes. (**A**) Immunoblot analysis of Cisd protein levels in wildtype and Cisd knockout (KO) flies, actin served as loading control. Note the presence of Cisd monomers (M), dimers (D), and multimers (X). (**B**) Quantitative PCR (qPCR) analysis of *Pink1* mRNA levels. *RpL32/Rp49* served as housekeeping control. Each dot represents the mean relative transcriptional level of a sample of five flies. The values were normalized to the mean value of wildtype flies. (**C**) Climbing ability. DKO, double-knockout flies. Each dot represents the percentage of a group of 25 flies that climbed more than 8 cm within 10 s. (**D**) Abnormal wing posture evaluation. Percentage of flies with an abnormal wing posture. Each dot represents an individual trial of a group of 25 flies. (**E**) Median survival on standard food. Fresh food was supplied and the number of dead flies was scored every 2 days. Each dot represents the median lifespan of a group of 25 flies. (**F**) Dopamine levels quantified by HPLC. Each dot represents a group of 4 flies sampled at different days. (**G**) Representative transmission electron microscopy of flight muscle morphology from Pink1 mutant and DKO. Scale bar as indicated. Exemplary pictures from wildtype and *Cisd* KO flies are shown in *Figure 5—figure supplement 4*. (**H**) Quantification of the indicated parameters. Each dot represents the average values from one image from 2 flies per genotype that were used for the analysis. Data are presented as box and whisker plots with the box extending from the 25th to the 75th percentile and the whiskers from the minimum to the maximum value depicting all individual data points. Statistical analysis was done using one-way ANOVA, p values are indicated.

The online version of this article includes the following source data and figure supplement(s) for figure 5:

**Source data 1.** Raw and labeled blots of wildtype and Cisd knockout flies probed against CISD1 and Actin.

**Source data 2.** Raw data of quantifications of climbing and wing posture and survival and dopamine levels and mitochondrial EM parameters of wildtype and CisdKO and Pink1B9 and doubleknockout flies.

**Figure supplement 1.** RNAi-mediated knockdown of Cisd protects against Pink1 loss of function.

**Figure supplement 1—source data 1.** Raw and labeled blots showing Cisd knockdown in flies.

**Figure supplement 1—source data 2.** Excel file containing raw data of the quantification of the blot showing Cisd KD as well as survival curve of Pink1B9 hemizygous flies.

**Figure supplement 2.** Phenotypes of Cisd knockout (KO) flies.

**Figure supplement 2—source data 1.** Raw data from climbing and survival of wildtype and CisdKO flies.

**Figure supplement 3.** Double-knockout flies lack Cisd protein expression.

**Figure supplement 3—source data 1.** Raw and labeled blots against CISD1 in fly strains wildtype and CisdKO and Pink and Park mutant flies and double knockout flies.

**Figure supplement 3—source data 2.** Raw and labeled blots of indicated fly strains blotted against CISD1.

**Figure supplement 4.** Exemplary transmission electron microscopy pictures.

**Figure supplement 4—source data 1.** Labeled image of transmission electron microscopy from flies' flight muscle of indicated strains.

**Figure supplement 4—source data 2.** Raw transmission electron microscopy images with scale bar from flight muscle of indicated strains.

generated double-knockout flies (DKO) lacking both *Cisd* and *Pink1*, with the loss of Cisd expression shown in *Figure 5—figure supplement 3*. Remarkably, when compared to *Pink1[B9]* flies these DKO exhibited normal climbing ability, normal wing morphology, normal dopamine levels, and a normal lifespan, essentially representing a complete rescue of these phenotypes (*Figure 5C–F*).

We further investigated whether these favorable effects of *Cisd* KO directly influenced mitochondria or stemmed from alternative non-mitochondrial factors. Flight muscles are densely populated with mitochondria and have been previously shown to be significantly impacted by the *Pink1[B9]* mutation (*Clark et al., 2006*; *Park et al., 2006*). In *Pink1[B9]* flies, flight muscles are atrophied and mitochondria cover less space, are smaller, and have a significantly reduced number of cristae resulting in a reduction of cristae surface (*Figure 5G*, a comparison of all four genotypes is shown in *Figure 5—figure supplement 4*). Intriguingly, the concurrent *Cisd* KO effectively rescued all these phenotypes (*Figure 5H*). Together these data strongly suggest that Cisd, most probably apo-Cisd, assumes a toxic function in *Pink1* mutant flies and serves as a critical mediator of Pink1-linked phenotypes, including the disruption of the normal mitochondria and cristae morphology.

### *Cisd* gene reduction partially protects *Prkn* mutant flies

*Pink1[B9]* and *Prkn* mutant flies (named *Park[25]*, also w[1118] background; *Park et al., 2006*) exhibit remarkably similar phenotypes, and previous elegant work in flies established that *Pink1* functions upstream of *Prkn* (*Clark et al., 2006*; *Park et al., 2006*). Furthermore, Cisd is the most prominent *Prkn* target in flies (*Martinez et al., 2017*). We therefore next investigated whether *Prkn* mutation also affects Cisd abundance or dimerization and whether loss of *Cisd* also protects against *Prkn* loss-of-function pathology. Given the absence of functional antibodies against fly Prkn, we used quantitative PCR (qPCR) to verify *Prkn* depletion in *Prkn* mutant flies. As expected, *Park[25]* flies expressed no *Prkn* mRNA, but, intriguingly, *Cisd* mRNA was also dramatically reduced in Prkn mutant flies (*Figure 6A*). Similarly, *Prkn* mRNA was nearly undetectable in *Cisd* KO flies (*Figure 6B*). When we examined total Cisd protein levels, we observed a downregulation of total Cisd levels in both young and old *Prkn* mutant flies in line with the mRNA results (*Figure 6C*). An exemplary blot is provided in *Figure 6C'*, a blot demonstrating Cisd depletion also in Prkn/Cisd DKO is shown in *Figure 5—figure supplement 3*. The dimer/monomer ratio was unchanged and displayed again a considerable degree of variability (*Figure 6D, D'*). We concluded that the substantial regulation of *Prkn* mRNA in *Cisd* KO flies and *Cisd* mRNA and protein in *Park[25]* flies suggests that these genes function within related pathways.

Like *Pink1[B9]* flies, *Park[25]* flies exhibited a notable reduction in climbing ability (*Figure 6E*), an abnormal wing posture (*Figure 6F*), and a shortened lifespan (*Figure 6G*), consistent with previous reports (*Clark et al., 2006*; *Park et al., 2006*). Intriguingly, concurrent Cisd KO conferred complete protection against the climbing impairment of *Park[25]* flies (*Figure 6E*) and showed a distinct inclination toward normal wing posture with DKO being not significantly different from WT flies (*Figure 6F*). However, surprisingly, DKO did not rescue the lifespan of *Park[25]* flies (*Figure 6G*). This observation suggests that Cisd likely serves an essential, yet undiscovered, function, particularly in the aging fly, and its loss is compensated for in the presence of Prkn.

## Discussion

We here identified the mitochondrial protein CISD1, specifically its iron-depleted form, as a potential downstream mediator of the pathophysiological effects arising from the loss of PINK1 function, and to a certain extent, Parkin loss of function.

Our investigation revealed that both dopaminergic neurons from familial PD patients with *PINK1* mutations and Pink1 mutant flies exhibit increased levels of a slower-migrating form of CISD1/Cisd on immunoblots. The presence of this slow migrating form of CISD1/Cisd was age-dependent in flies and apparent in human dopaminergic neurons after 28 days of differentiation. We did not assess whether this increased even more in older cultures but such an age-dependent increase would be in line with the fact that PD usually becomes symptomatic at an older age. The slower-migrating form likely does not represent a ubiquitinated form, as treatment with the deubiquitinase USP2 had no effect on fly Cisd dimerization (*Martinez et al., 2024*). Additionally, it is improbable that the slower-migrating Cisd form is an unrelated protein, as under non-reducing conditions, fly Cisd shifted to the dimeric form (*Martinez et al., 2024*) while harsh reducing conditions shifted Cisd to the monomeric

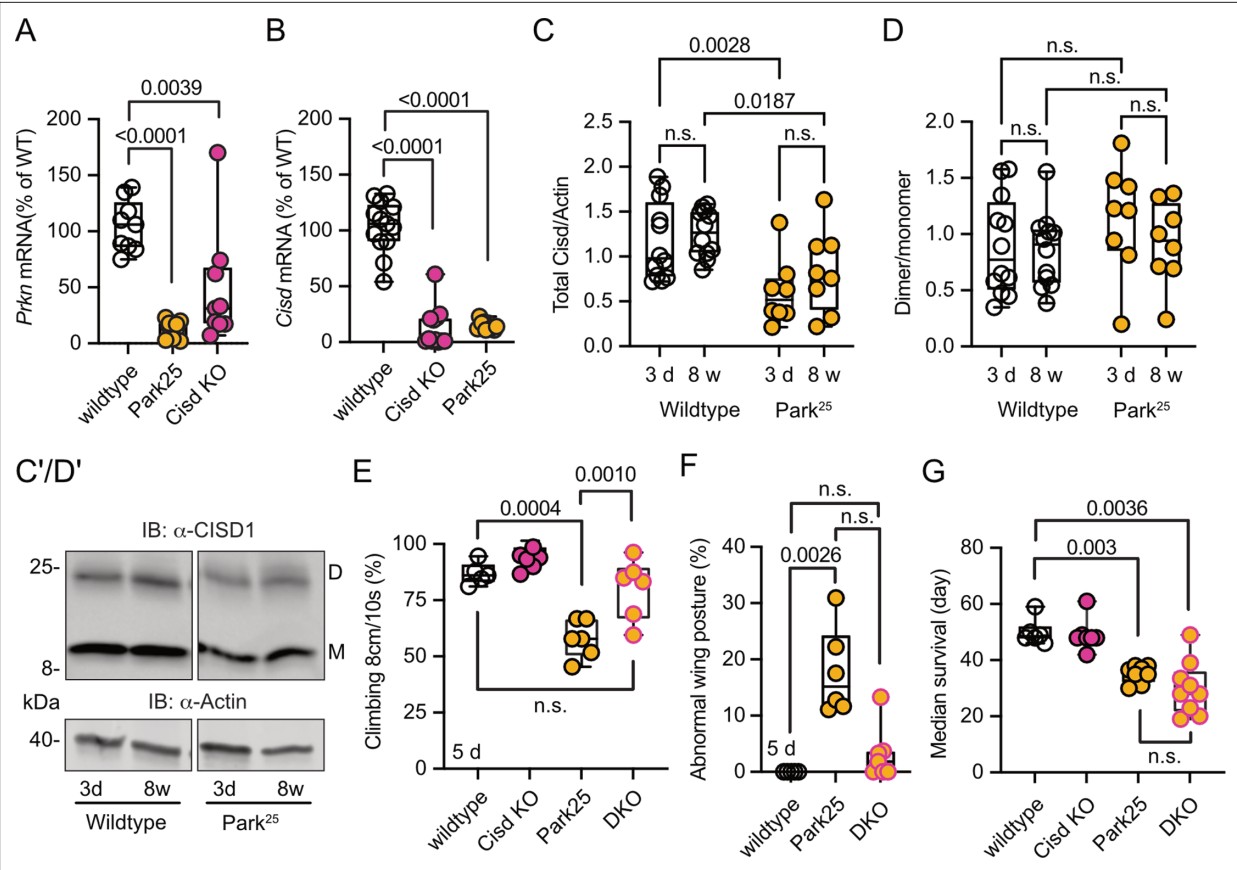

**Figure 6.** *Cisd* gene reduction partially protects *Prkn* mutant flies. (**A, B**) Quantitative PCR analysis of *Prkn* and *Cisd* mRNA levels in *Cisd* knockout (KO) and Park25 flies. *RpL32/Rp49* served as housekeeping control. Each dot represents the relative transcriptional level of a sample of five flies. The values were normalized to the mean value of wildtype flies. (**C**) Total Cisd (D+M) normalized to actin. Each dot represents the mean of two technical replicates from n=12 wildtype and n=8 Park25 independent samples of two male flies. The wildtype samples shown for comparison are the same as in *Figure 5C/D*. (**D**) Dimer/monomer ratio of the same samples. (**C', D'**) Immunoblot analysis of Cisd abundance in the indicated fly strains at different ages, 3 days (d) and 8 weeks (w). Actin served as loading control, size is indicated. (M) Cisd monomers and (D) dimers. The gap indicates the removal of the Pink1[B9] samples shown in *Figure 5* that were run on the same blot. (**E**) Climbing ability assessment. Each dot represents the percentage of a group of 25 flies that successfully climbed more than 8 cm within 10 s. (**F**) Abnormal wing posture evaluation. Percentage of flies with an abnormal wing posture. Each dot represents an individual trial of a group of 25 flies. (**G**) Median survival on standard food. Fresh food was supplied and the number of dead flies was scored every 2 days. Each dot represents the median lifespan of a group of 25 flies. Data are presented as box and whisker plots with the box extending from the 25th to the 75th percentile and the whiskers from the minimum to the maximum value depicting all individual data points. Statistical analysis was done using one-way ANOVA in A/B/F/G and two-way ANOVA in C/D, p values are indicated.

The online version of this article includes the following source data for figure 6:

**Source data 1.** Raw and labeled blots of CISD1 expression in wildtype and Park25 flies.

**Source data 2.** Raw data of the quantification of quantitative PCR analysis and immunoblot analysis and climbing assay and wing posture and survival of wildtype and Park25 and dKO flies.

form (*Figure 3E*). Based on our experiments, we conclude that the slower-migrating form of CISD1/Cisd corresponds to a dimer caused by the formation of an intermolecular disulfide bond between two molecules of CISD1/Cisd. The dimeric form has been described before even in recombinant CISD1. Under non-reducing conditions, recombinant CISD1 ran as a monomeric and a dimeric band while under reducing conditions using beta-mercaptoethanol only the monomeric band was detected (*Roberts et al., 2013*). Using proteomic analysis, it was determined that the disulfide bond forms between cysteine C83 and either C72 or C74 of CISD1 (*Roberts et al., 2013*). Formation of the disulfide bond therefore results in loss of the Fe/S cluster because the residues involved in bond formation also coordinate the Fe/S cluster of CISD1. It is unclear what comes first, loss of the Fe/S cluster followed by disulfide bond formation or vice versa disulfide bond formation followed by loss of the Fe/S cluster. However, CISD1 C83S, a genetically engineered CISD1 incapable of binding the

iron-sulfur cluster and probably also incapable of forming a disulfide bond (*Roberts et al., 2013*), has a higher propensity for dimer formation when analyzed in a cellular assay based on complementation of split luciferase (*Figure 2A*). This suggests that Fe/S cluster loss comes first and results in an altered structure that facilitates the formation of the disulfide bond. The iron-sulfur cluster of CISD1 is easily lost because it is coordinated in an unconventional manner by one histidine (H) and three cysteines (C) instead of the more common 2H/2C or 4C coordination (*Wiley et al., 2007b*). This serves to transfer the iron-sulfur cluster to other proteins as part of a cellular adaptive response that helps recovery from oxidative injury (reviewed in *Mittler et al., 2019*).

One protein that can accept the Fe/S cluster from holo-CISD1 is IRP1, as previously demonstrated (*Ferecatu et al., 2014*). Also in flies, Cisd physically interacts with holo-IRP1, and its deficiency renders flies more susceptible to iron depletion (*Huynh et al., 2019*), as also observed by us for *Pink1* mutant flies. How apo-CISD1 is then reconstituted is rather unclear. A recent report suggested that CISD1 can receive its Fe/S cluster from its homolog CISD3 (aka MiNT) in the mitochondrial matrix via the porin VDAC1 (*Karmi et al., 2022*). So it appears possible that either this transfer is inhibited/attenuated or that the apo-CISD1/Cisd observed in *PINK1* mutant dopaminergic neurons and *Pink1* mutant flies constitutes a permanently fixed apo-CISD1/Cisd that cannot be reconstituted. This disruption could result in a defective recycling of cytosolic apo-IRP1 into holo-IRP1, leading on the one hand to a deficiency in cytosolic aconitase activity. Knockdown of CISD1 in cells indeed significantly reduces the cytosolic aconitase activity (*Tan et al., 2016*). On the other hand, apo-IRP1 not only lacks aconitase activity but also binds to IREs in the mRNA of iron-responsive genes (*Haile et al., 1992a*; *Haile et al., 1992b*). Association with IREs in the 5' untranslated region of ferritin mRNA inhibits ferritin translation resulting in less bound iron and an increase in free iron (*Chen et al., 1997*) while association with IREs in the 3' untranslated region of the mRNA encoding the iron importers transferrin receptor 1 and SLC11A2 stabilizes the mRNAs and facilitates their translation resulting in increased iron import (*Galy et al., 2010*). Together the transcriptional changes caused by the presence of apo-IRP1 result in iron overload which is an important factor in the pathology associated with PD (recently reviewed in *Ma et al., 2021*). Our study also revealed increased total iron levels in Pink1 mutant flies. Notably, KO of PINK1 in flies (*Esposito et al., 2013*), in mice (*Gautier et al., 2008*), and in human dopaminergic neurons (*Bus et al., 2020*) also results in a reduced aconitase activity in mitochondria. Mitochondrial aconitase contains an IRE in its 5' untranslated region (*Kim et al., 1996*), making its expression subject to repression by apo-IRP1. The inactivation of mitochondrial aconitase caused by Pink1 loss of function triggers increased superoxide production, oxidative stress, and mitochondrial swelling (*Esposito et al., 2013*). Others reproduced these findings and reported that overexpression of mitoferrin, a mitochondrial iron transporter, mitigates the reduced mitochondrial aconitase activity, abnormal wing posture, flight deficits, and mitochondrial morphology defects associated with Pink1 loss of function by elevating mitochondrial bioavailable iron levels (*Wan et al., 2020*). Interestingly, KO of IRP2, a protein highly homologous to IRP1 but lacking aconitase activity, results in iron accumulation and Parkinsonism in mice (*LaVaute et al., 2001*). An interaction of CISD1 with IRP2 has not been reported so far but appears likely. In summary, these results strongly implicate altered mitochondrial iron homeostasis and redox dysregulation in fly and human models of PD and link it to CISD1 dysfunction. The accumulation of apo-CISD1/Cisd and the subsequent inability to repair defective IRP1/2 would place CISD1 upstream of the changes in iron homeostasis observed in PD and models of PD (*Ma et al., 2021*).

Our results also suggest that apo-CISD1 exhibits a toxic gain of function possibly independent of the effects on iron homeostasis. Ubiquitous overexpression of Cisd and apo-Cisd in flies aggravated known phenotypes of *Pink1* mutation as a reduced climbing ability and lifespan. This was evident in WT and *Pink1* mutant flies and overexpression of apo-Cisd was associated with a more profound reduction in climbing speed and resulted in an inability to completely inflate their wings. Interestingly, prolonged exposure of MEFs to the iron-depleting agent 2,2'-bipyridine results in downregulation of Cisd1 (*Key et al., 2020*) possibly as a compensatory mechanism to limit the abundance of such a toxic iron-depleted CISD1. In line with a toxic function of Cisd and apo-Cisd, removing fly Cisd by KO of the coding sequence completely protected flies against *Pink1* depletion-induced changes in climbing ability, abnormal wing posture, dopamine levels, and lifespan but ameliorated only the climbing ability and wing posture in Prkn mutant flies. Previous work demonstrated that ectopic overexpression of Cisd in muscle decreased the lifespan of flies and resulted in enlarged mitochondria

containing numerous intramitochondrial vesicles surrounded by a double membrane (*Chen et al., 2020*). Our data now indicate that this process might contribute to the pathophysiology resulting from the loss of Pink1/Prkn-mediated mitochondrial quality control.

Very recent work, published during the revision of this manuscript, discovered that both genetic and pharmacological suppression of CISD1 and Cisd restored increased calcium release from the ER conferred by *PINK1* and *Parkin* mutation in mammalian cells and flies (*Ham et al., 2023*). However, the specific role of apo-Cisd in this remains to be fully elucidated. Also Martinez et al. very recently reported that *Cisd* depletion using RNAi protects against *Pink1* and *Prkn* loss of function in flies (*Martinez et al., 2024*), suggesting that a reduction, instead of a complete KO, of *Cisd* as used by us is more beneficial in *Park25* flies and equally beneficial in *Pink1B9* flies. In their hands, Cisd levels increased in *Pink1* and *Prkn* mutant flies which was not evident in our study. Cisd overexpression was neurotoxic, disrupted mitochondrial morphology, and led to an accumulation of phospho-ubiquitin which they speculate also blocks mitophagy, resulting in the accumulation of more defective mitochondria (*Martinez et al., 2024*). All three papers propose distinct mechanisms for the toxic effects of CISD1 in PD. While our study suggests a role in redox imbalance and iron dysregulation, particularly implicating the iron-depleted form of CISD1, *Ham et al., 2023*, describe its impact on ER calcium release, and *Martinez et al., 2024*, propose interference with autophagy/mitophagy. These mechanisms are not mutually exclusive, as they often intersect. Both studies support our findings, replicating key results, but also underscore the need for further research to fully elucidate the interplay of these proposed mechanisms.

In summary, our results suggest that the mitochondrial iron-sulfur cluster protein Cisd is downstream of the pathophysiological cascade initiated by Pink1 and partially Prkn loss of function in flies and probably also in humans. This sheds light on the pathophysiology of PD and implicates CISD1 as a therapeutic target protein.

## Methods

### Human dopaminergic neurons

iPSCs from two PD patients carrying the p.Q456X mutation in PINK1 were obtained from the University of Lübeck. Both participants signed a written informed consent according to the Declaration of Helsinki. Ethical approval for conducting iPSC studies in Krüger's lab was granted by the National Committee for Ethics in Research, Luxembourg (Comité National d'Ethique de Recherche; CNER #201411/05). For both mutant lines, isogenic gene-corrected controls have been successfully generated and characterized previously (*Jarazo et al., 2022*). Both PINK1-mutant and isogenic iPSCs were first converted into NESCs, based on the protocol previously described (*Reinhardt et al., 2013*). NESCs were cultivated in N2B27 medium (DMEM-F12/Neurobasal 50:50, supplemented with 1:200 N2, 1:100 B27 without vitamin A, 1% L-glutamine and 1% penicillin/streptomycin), containing 3 µM CHIR-99021, 0.5 µM purmorphamine (PMA), and 150 µM ascorbic acid. For the generation of midbrain-specific neurons, NESC maintenance medium was replaced by N2B27 containing 1 µM PMA, 200 µM ascorbic acid, 10 ng/mL BDNF, 10 ng/mL GDNF, 1 ng/mL TGF-b3, and 500 µM dbcAMP. After 6 days, differentiating NESCs were shifted to the same medium but without PMA (replaced every 2 days) and kept until day 28.

### Immunoblotting

Cells were lysed in RIPA buffer (25 mM Tris•HCl pH 7.6, 150 mM NaCl, 1% NP-40, 1% sodium deoxycholate, 0.1% SDS) containing protease and phosphatase inhibitors. Cells were then centrifuged at maximum speed for 30 min at 4°C and the supernatant containing the whole-cell proteins was collected. Flies were isolated in the same buffer but were mechanically smashed with a dounce, centrifuged and the supernatant was collected. For regular immunoblots, protein samples were denatured in 1× loading Laemmli sample buffer (Tris-HCl, SDS, glycerol, bromophenol blue) already containing 50 mM DTT, samples for non-reducing blots were denatured in 1× loading Laemmli sample buffer without DTT. Samples were then boiled at 95°C for 5 min. Harsher reducing conditions were generated by treating the samples with 100 mM DTT or the combination of 100 mM DTT and 50 mM TCEP, incubation for 30 min at RT, and boiling for 10 min at 95°C. The Cisd redox blot was used to quantify the ratio between monomeric and dimeric Cisd following the alkylation of reduced thiols.

Here, 100 mM NEM was added to the lysis buffer to preserve the in vivo redox state of Cisd and to prevent redox changes during sample preparation. Proteins were separated by gel electrophoresis and transferred to nitrocellulose membranes using a semi-dry blotting system (Bio-Rad Laboratories). Membranes were blocked with 3% milk or 5% BSA in TBST for 1 hr at RT. Membranes were incubated with the respective antibodies overnight at 4°C, rotating. Antibodies used were rabbit anti-CISD1 (Proteintech, 1:1000), mouse anti-actin (Merck Chemicals, 1:1000), rabbit anti-HA (Abcam, 1:1000), rabbit anti-TH (Millipore, AB152, 1:1000), mouse anti-TUBB3 (Bio-Legend, 801201, 1:20,000), and mouse anti-β-actin (Cell Signaling Technology, 3700, 1:30,000). Fluorescence-labeled secondary antibodies were used at a 1:10,000 dilution, and the signal was detected using a Li-Cor Odyssey imaging system and quantified with the Image Studio Lite software. This imaging system is linear and oversaturation is indicated. The intensity was normalized to the loading control and the mean per blot.

## Dimerization assay

Split nanoluc fragments LgBit and SmBit (Promega) were cloned N-terminally to human CISD1 and point mutations introduced by site-directed mutagenesis using the Q5 Site-Directed Mutagenesis kit (NEB). All plasmids were sequence verified. Experiments were carried out by transfecting HT22 cells with the plasmids for 48 hr after which 1 volume of the Nanoluc Live-Cell substrate was added with 19 volumes of Nano-Glo LCS Dilution Buffer. Where indicated, a final concentration of FCCP (1 µM), oligomycin/antimycin (2.5 µM each), deferiprone (1 mM; *Allen et al., 2013*) or vehicle was added to the cells 2 hr prior to the addition of the substrate.

## Fly experiments

Cisd KO flies were generated by Wellgenetics (Taiwan, ROC) and contain the cassette depicted in *Figure 5—figure supplement 2A*. *Pink1[B9]*, *Park[25]*, *UAS-mtGFP*, *UAS-Cisd-WT*, *UAS-Cisd C/S* flies were a kind gift of Alex Whitworth (Mitochondrial Biology Unit, MRC, Cambridge, UK). *TubP-Gal4* (BDSC 5138), *Da-Gal4* (BDSC 8641), *Elav-Gal4* (BDSC 458), and *UAS-Cisd2-RNAi* (BDSC 33749) were obtained from the Bloomington Stock Center (BDSC). The RNAi control strains *UAS-vasa-RNAi* (VDRC46464) and *UAS-always-early-RNAi* (VDRC13673) were from the Vienna *Drosophila* Resource Center (VDRC). *Lifespan:* Fruit flies were provided with standard molasses-based food and accommodated in a climate chamber at a temperature of 25°C with a 12 hr light and 12 hr darkness schedule. The flies were given fresh food every 2 days and dead flies were scored. Each experiment consisted of four groups of 25 flies. Where indicated, fly food was prepared with deferiprone with a final concentration of 65 µM. *Climbing:* A total of four groups, each consisting of 25 flies per genotype per age, were tested in vials of a 2 cm diameter. The percentage of flies that were able to successfully traverse a height of 8 cm within 10 s or 8 s (as indicated) was documented. The assay was always replicated at the same time of day. *Scoring for eclosed flies:* Fly food was prepared either with vehicle or with deferiprone (65 µM). A 3:1 ratio of female virgin flies to male flies (5 days of age) were added and allowed to mate. These were then removed and the number of eclosing male flies per vial was quantified each day for 3 days. *Scoring wing phenotype:* A 3:1 ratio of female virgin flies to male flies (5 days of age) were added and allowed to mate. These were then removed and the number of flies eclosing with curly wings or non-inflated wings was noted for 3 days. *Quantitative RT-PCR:* RNA extraction was carried out with the ZR RNA MiniPrep kit (ZYMO RESEARCH) and cDNA was synthesized from 10 ng/µL RNA using the High Capacity cDNA Reverse Transcription Kits (Life Tech). FastStart Universal SYBR Green Master (Rox) (Merck) was used for qPCR, with primers from Eurofins available on demand. The $2-\Delta\Delta Ct$ (Ct, cycle of threshold) method was employed to calculate transcriptional levels, where $\Delta\Delta Ct = \Delta Ct$ of experimental group − mean $\Delta Ct$ of control groups. $\Delta Ct =$ Ct (gene of interest) − Ct (RpL32/Rp49). RpL32/Rp49 served as housekeeping control. All flies used for these experiments were male. *Genomic DNA isolation and PCR:* Genomic DNA was extracted from flies using the Biozym gDNA Kit and PCR amplification performed to assess the deletion in the Pink1 gene using the Q5 High-Fidelity DNA Polymerase (New England Biolabs). *Primer sequences: rp49* F 5′FCGGATCGATATGCTAAGCTGT 3′ R GCGCTTGTTCGATCCGTA 3′. *Cisd* F 5′GGCCAACC ATCCTTGAA 3′R 5′ CTTGTCAAGTCTGCCGGTTC 3′. *Pink* F 5′CTCTGTGCACAACGTCTTCCACTA CAG 3′ R 5′GAATTCGCTGGAGAAGTCACATTGAAG 3′. *GstS1* F 5′ATCAAGCACAGCTACACGCT 3′ R 5′ CGAGGAATCGAGCCATCGAA 3′.

## Dopamine quantification

Flies were anesthetized with $CO_2$ and heads removed while gently holding the flies with a tweezer. Heads were immediately put in a solution containing 200 µL 5% TCA, 300 µL RIPA buffer, and 10 µL dopamine standards at 4°C. Ceramic beads were added and lysis was performed by gently shaking for 5 min. Lysates were cleared by centrifugation for 5 min at 13,000 rpm twice. 200 µL of the supernatant was put in a fresh vial. Dopamine was determined via HPLC/MS/MS. 100 µL of 100 mM ammonium formate buffer (adjusted to pH 2.5 with formic acid) were added to 100 µL of the prepared extracts from fly heads in order to adjust pH in an HPLC vial with insert. 50 µL of this solution were injected into the LC/MS/MS system consisting of an Agilent 1260 Infinity II Series HPLC pump (G7112B) with integrated degasser and an Agilent 1260 Autosampler (G7167A) connected to an AB Sciex 4500 tandem mass spectrometer. Chromatography was performed on a Phenomenex Kinetex F5 column (2.6 µM, 150 mm × 4.6 mm) using water (adjusted to pH 2.5) and acetonitrile as solvents A and B, respectively. Starting conditions were 5% solvent B for 1 min, raised to 100% B at 10 min, kept at 100% B until 14 min and set back to 5% B until 19 min and kept for another 5 min until 24 min. Flow rate was 0.3 mL/min. Tandem mass spectrometric determination was performed in ESI-negative mode using the specific ion transition 154.0→>137.0 for dopamine and 158.0→140.9 for D4 dopamine as internal standard. We used an electrospray needle voltage of – 4500 V and nitrogen as a nebulizer and turbo heater gas (450°C) was set to a pressure of 65 psi. Nitrogen as curtain gas and collision gas was set to 10 instrument units, respectively. Calibration was performed in a mixture of 300 µL RIPA buffer and 200 µL 5% trichloroacetic acid in water, spiked with dopamine concentrations ranging from 1 to 50 µg/L in HPLC vials. 10 µL of the working solution of the internal standard ($D_4$-dopamine, 1 mg/L in MeOH/HCl) were added to the standards, the vials were vortexed and centrifuged at 4.000 U/min (~3000×$g$). 100 µL of the supernatant was pipetted in a new HPLC vial with insert and 100 µL of 100 mM ammonium formate buffer (pH 2.5) were added. 50 µL of this solution was injected into the LC/MS/MS system as described. As dopamine is not stable under neutral/basic conditions, the spiking solutions of dopamine (1 mg/L and 100 µg/L) were freshly prepared daily from the stock solution (1 g/L in MeOH/HCl) in 0.1% $HCl_{konz.}$ in water.

## Iron quantification

Iron redox speciation in fly lysates was carried out using CE-ICP-MS. To ensure interference-free monitoring of iron isotopes, dynamic reaction cell (DRC) technology employing $NH_3$ as the DRC gas was utilized. A 'PrinCe 706' CE system (PrinCe Technologies B.V., Emmen, Netherlands) was employed for the separation process, applying a voltage of +20 kV. The temperature settings for both the sample/buffer tray and capillary were maintained at 20°C. An uncoated capillary (100 cm×50 µm ID; CS-Chromatographie Service GmbH, Langerwehe, Germany) was used for interfacing with ICP-DRC-MS. The CE-ICP-MS interface enabled electrical connection while preventing suction flow by operating in self-aspiration mode (*Michalke et al., 2019*). The electrolytes used included 10% HCl as the leading electrolyte, 0.05 mM HCl as the terminating electrolyte, and 50 mM HCl as the background electrolyte. The $Fe^{2+}/Fe^{3+}$ ratio was determined based on quantified concentrations of Fe species. An uncoated capillary (CS-Chromatographie Service GmbH, Langerwehe, Germany) measuring 85 cm in length with a 50 µm internal diameter was employed, connecting the CE system to the ICP-MS, served as the element-selective detector. A 'Nexlon 300-D' ICP-DRC-MS system (Perkin Elmer, Rodgau-Jügesheim, Germany) served as the element-selective detector for iron electropherograms. Isotopes $^{56}Fe$ and $^{57}Fe$ were measured in DRC mode using ammonia as the DRC gas (0.58 mL $NH_3$/min). The radio frequency (RF) power was set to 1250 W, and the plasma gas flow was 16 L Ar/min. The nebulizer gas flow rate at the CE-ICP-DRC-MS interface was optimized to 0.98 L Ar/min. Electropherograms were analyzed using Peakfit software (Systat, Inpixon GmbH Office, Düsseldorf, Germany). To assess iron recovery accuracy, total iron content was determined using ICP-sector field mass spectrometry (ICP-sf-MS). The ICP-sf-MS ('ELEMENT 2', Thermo Scientific, Bremen, Germany) settings included an RF power of 1260 W, a plasma gas flow of 16 L Ar/min, an auxiliary gas flow of 0.85 L Ar/min, a nebulizer gas flow of 1.085 L Ar/min, and a dwell time of 300 ms. Quantified iron species from CE-ICP-DRC-MS were compared to total iron content (set at 100%), yielding values ranging between 92% and 110%.

## Transmission electron microscopy

Fly thoraxes were isolated and fixed in a buffer containing Caco-buffer (0.2 M), 25% glutaraldehyde, and 15% paraformaldehyde. After fixation, tissues were washed in a wash buffer (0.2 M Caco-buffer diluted 1:1 with ddH$_2$O) and incubated with 2% osmium in 0.2 M buffer. Tissues were washed again and dehydrated in ascending ethanol series 30–70% then left overnight at 4°C. The next day, the samples were dehydrated further in 80%, 90%, 95%, and 100% ethanol solutions then washed with propylenoxide and left overnight in a 1:1 dilution of resin:proyleneoxide. Samples were moved the next day to just resin. The last day, flies were placed into a special mold with resin and left in an oven for 2 days. Samples were then cut and contrasted. Images were acquired with a Tecnai 12 Transmission Electron Microscope.

## Statistical analysis

Statistical analysis was performed using GraphPad Prism. Data were tested for normality using the Shapiro-Wilk test. Two parametric samples were compared using the Student's t test, two nonparametric samples using the Mann-Whitney test, two normalized parametric samples using the one-sided t test, two nonparametric samples using the Wilcoxon signed-rank test. More than two parametric samples were compared using one-way or two-way ANOVA and nonparametric samples using the Kruskal-Wallis test followed by the respective post hoc tests.

## Acknowledgements

We thank Christine Klein and Philipp Seibler (University of Lübeck, Germany) who kindly shared the human iPSC lines from PD patients carrying the PINK1 p.Q456X mutation, and Jens Schwamborn and Javier Jarazo (LCSB, University of Luxembourg, Luxembourg) for providing the corresponding gene-corrected controls. We also thank Marion Silies, JGU Mainz, for providing lab space and helpful discussions. This work contains data from the medical theses of Timo Baumann, Majd Abusaada, and Christopher Weber.

## Additional information

### Funding

| Funder | Grant reference number | Author |
|---|---|---|
| Deutsche Forschungsgemeinschaft | 445683311 | Axel Methner |
| Deutsche Forschungsgemeinschaft | 461705066 | Vivek Venkataramani Axel Methner |
| Fonds National de la Recherche Luxembourg | C21/BM/15850547/PINK1-DiaPDs | Giuseppe Arena |

The funders had no role in study design, data collection and interpretation, or the decision to submit the work for publication.

### Author contributions

Sara Bitar, Formal analysis, Investigation, Methodology, Writing - review and editing; Timo Baumann, Majd Abusaada, Liliana Rojas-Charry, Patrick Ziegler, Thomas Schettgen, Isabella Eva Randerath, Bernhard Michalke, Investigation; Christopher Weber, Investigation, Writing - review and editing; Vivek Venkataramani, Supervision, Funding acquisition, Investigation, Writing - review and editing; Eva-Maria Hanschmann, Conceptualization, Resources, Formal analysis, Writing - review and editing; Giuseppe Arena, Funding acquisition, Investigation, Writing - review and editing; Rejko Krueger, Supervision, Project administration; Li Zhang, Supervision, Writing - review and editing; Axel Methner, Conceptualization, Data curation, Formal analysis, Supervision, Funding acquisition, Validation, Visualization, Methodology, Writing - original draft, Project administration, Writing - review and editing

## Author ORCIDs
Sara Bitar ⦿ http://orcid.org/0000-0002-9369-0500
Timo Baumann ⦿ https://orcid.org/0000-0001-5374-380X
Thomas Schettgen ⦿ https://orcid.org/0000-0003-1256-1713
Giuseppe Arena ⦿ https://orcid.org/0000-0003-2398-5503
Li Zhang ⦿ https://orcid.org/0000-0002-6339-6087
Axel Methner ⦿ https://orcid.org/0000-0002-8774-0057

## Ethics
Induced pluripotent stem cells (iPSCs) from two PD patients carrying the p.Q456X mutation in PINK1 were obtained from the University of Lübeck. Both participants signed a written informed consent according to the Declaration of Helsinki. Ethical approval for conducting iPSC studies in Krüger's lab was granted by the National Committee for Ethics in Research, Luxembourg (Comité; National d'Ethique de Recherche; CNER #201411/05).

## Decision letter and Author response
Decision letter https://doi.org/10.7554/eLife.97027.sa1
Author response https://doi.org/10.7554/eLife.97027.sa2

# Additional files

## Supplementary files
• MDAR checklist

## Data availability
All data generated or analysed during this study are included in the manuscript and supporting files.

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
