## [Editor Report]

The study focuses on the role of CISD1 in Parkinson's disease (PD) and its relationship with the PINK1/Parkin pathway. The obtained data provide convincing evidence that apo-CISD1 may have a toxic function in Pink1 mutant flies and in dopaminergic neurons from patients with PINK1 mutation, thus serving as a critical mediator of Pink1-linked PD phenotypes. The findings provide valuable insights into the role of iron homeostasis and mitochondrial biology in PD.

---

## [Decision Letter]

[Editors' note: this paper was reviewed by Review Commons.]

---

## [Author Response]

We thank the reviewers for their excellent work that greatly improved our work. We are very content that reviewer #1 considered our work to be “novel, interesting and important for understanding the mitochondrial biology of PD”. This reviewer also valued our work as “a significant advancement” and suggested further study of the relationship of CISD1 (dimerization) to general mitophagy/autophagy. We already addressed this in the first revision (version 1, v1).

Also reviewer #2 considered our work to be “an exciting and well-executed piece of research focusing on the defects in iron homeostasis observed in Parkinson's disease which a wide audience will appreciate”. This reviewer had a very specific suggestion on how to improve our manuscript which makes a lot of sense and is feasible. As the suggested experiments include fly breeding and behavioral analysis, these experiments took some time and are now included in the second revision (version 2, v2).

Finally, reviewer #3 gathered that parts of our results “are confirmatory to recently published work” but also appreciated that our results established that iron-depleted apo-Cisd is an important determinant of toxicity which has not been shown before. I would like to comment here, that in contrast to the paper mentioned by this reviewer, our contribution includes data from dopaminergic neurons obtained from human patients suffering from familial Parkinson’s disease that demonstrate the same increase in apo-CISD1 levels as the flies.

We also made additional changes in v2 not requested by the reviewers. New Figure 3E/F now demonstrates that the CISD1 dimer consists of two monomers that are covalently linked by a disulfide bridge. This is important because in this conformation CISD1 cannot coordinate the iron-sulfur cofactor. This work was done by an additional author, Eva-Maria Hanschmann (University of Essen, Germany), who is an expert in the field of redox modifications. We also decided to remove former Figure 3 showing the effect of apoCISD1 in mouse embryonic fibroblasts for clarity and brevity.

Reviewer #1 (Evidence, reproducibility and clarity (Required)):Summary:The manuscript focuses on mitochondrial CISD1 and its relationship to two Parkinson's disease (PD) proteins PINK1 and Parkin. Interestingly, CISD1 is a mitochondrial iron sulfur binding protein and an target of Parkin-mediated ubiquitinylation. Disruption of iron metabolism and accumulation of iron in the brain has long since been reported in PD but the involvement of iron sulfur binding is little studied both in vivo and in human stem cell models of PD. This work addresses the relationship between CISD1 and two mitochondrial models of PD (PINK1 and Parkin) making use of in vivo models (*Drosophila*), PINK1 patient models (iPSC derived neurons) and Mouse fibroblasts. The authors report a complex relationship between CISD1, PINK1 and Parkin, where iron-depleted CISD1 may illicit a toxic gain of function downstream of PINK1 and Parkin.Major comments:The conclusions are overall modest and supported by the data. One question remains unaddressed. Is mitochondrial CISD1 a downstream target that specifically mediates PINK1 and Parkin loss of function phenotypes or are the phenotypes being mediated because CISD1 is downstream of mitophagy in general?It would be interesting to know what happens to CISD1 (dimerization?) upon initiation of mitophagy in wild type cells? Would dissipation of mitochondrial membrane potential be sufficient to induce changes to CISD1 in wild type cells or PINK1 deficient cells? Since iron chelation is a potent inducer of mitophagy (Loss of iron triggers PINK1/Parkin-independent mitophagy. George F G Allen, Rachel Toth, John James, Ian G Ganley. EMBO Reports (2013)14:1127-1135) it would be useful to show one experiment addressing the role of CISD1 dimerization under mitochondrial depolarizing and non-depolarizing conditions in cells.

Based on the overall assumption of the reviewer that our work is “novel, interesting and important for understanding the mitochondrial biology of PD” and “a significant advancement” we understand the word “modest” here as meaning “not exaggerated”. To address this question, we studied CISD1 dimerization in response to more classical activators of mitophagy namely FCCP and antimycin/oligomycin which had no significant effect on dimerization suggesting that this phenotype is more pronounced under iron depletion. These data are shown in the new Figure 2C.

Alternatively, the authors should discuss the topic of mitophagy (including PINK1-parkin independent mitophagy), the limitation of the present study not being able to rule out a general mitophagy effect and previous work on the role of iron depletion on mitophagy induction in the manuscript.The data and the methods are presented in such a way that they can be reproduced.The experiments are adequately replicated and statistical analysis is adequate.Minor comments:Show p values even when not significant (ns) since even some of the significant findings are borderline < p0.05.

Here, I decided to leave it as it is, because the figures became very cluttered and less easy to understand. Borderline findings are however indicated and mentioned in the text.

Because the situation for CISD1 is complicated (overexpression, different models etc.) it would be helpful if in the abstract the authors could summarize the role. E.g. as in the discussion that iron-depleted CISD1 could represent a toxic function.

The abstract has been completely rewritten and now mentions the potential toxic function of iron-depleted CISD1.

If there is sufficient iron (accumulation in PD) why would CISD1 be deactivated? Perhaps that could be postulated or discussed in a simplified way?

We actually think that apo-CISD1 without its iron/sulfur cluster is incapable of transferring its Fe/S cluster to IRP1 and IRP2. This then results in increased levels of apo-IRP1/2 and subsequent changes that lead to iron overload. Such a sequence of events would place CISD1 upstream of the changes in iron homeostasis observed in PD and models of PD. This is now discussed in more detail.

In the methods section both reducing and non-reducing gel/Western blotting is mentioned but the manuscript only describes data from blots under reducing conditions. Are there blots under non-reducing conditions that could be shown to see how CISD1 and dimerized CISD1 resolve?

We now show blots where we used extremely reducing conditions of prolonged incubation with higher concentrations of DTT and tris-(2-carboxyethyl)-phosphine as a disulfide-bond breaking agent followed by boiling. This resulted in loss of dimeric and an increase of monomeric Cisd (new Figure 3E). We also added the thiol alkylating agent N-ethylmaleimide (NEM) prior and during lysis and performed non-reducing immunoblotting in old Pink1B9 flies compared to old wildtype flies. NEM blocks all reduced sulfhydryl (-SH) groups of cysteine residues and thereby prevents the formation of disulfide bonds due to non-specific oxidation during the lysis process (new Figure 3F). We also observed the increased dimer formation in old Pink1B9 flies under these conditions (new Figure 3F). Together these experiments indicate that the increased dimerization in old Pink1B9 flies is most probably caused by the formation of an intermolecular disulfide linking two Cisd molecules.

In the Results section, PINK1 mutant flies, it is said that the alterations to CISD1 (dimerization) are analogous to the PINK1 mutation patient neurons. The effect is seen in old but not young flies. Since iPSC-derived neurons are relatively young in the dish, would one not expect that young flies and iPSC-derived neurons have similar CISD1 phenotypes? Could the authors modify the text to reflect that? or discuss the finding in further context.

We only studied one time point in PINK1 mutation patient neurons and controls. It would indeed be interesting whether neuronal aging (as far as this can be studied in the dish) would result in increased CISD1 dimerization. This is now discussed.

Reviewer #1 (Significance (Required)):The strengths of this work are in the novelty of the topic and the use of several well established in vivo and cell models including patient-derived neurons. The findings discussed in the text are honest and avoid over-interpretation. The findings are novel, interesting and important for understanding the mitochondrial biology of PD.

We thank the reviewer for their kind words.

Limitations include the lack of strong phenotypes in the CISD1 models and the lack of robust, sustained and consistent increase in CISD1 dimers in the patient and fly models (just significant because of variability). The relationship of CISD1 (dimerization) to general mitophagy/autophagy is not shown here.

The lack of a strong phenotype in Cisd-deficient flies could actually hint to a potential compensatory mechanism that could also protect the Pink1 mutant x Cisd-deficient double-knockout flies. It is correct that the increase in CISD1/Cisd dimers in the PD models are not overwhelming but – as also mentioned by the reviewer – this could be increased in “older” cultures. This is discussed in more detail. We have now reproduced the increased dimerization of Cisd in samples where we added the thiol alkylating agent N-ethylmaleimide (NEM) prior and during lysis and performed non-reducing immunoblotting in old Pink1B9 flies compared to old wildtype flies. NEM blocks all reduced sulfhydryl (-SH) groups of cysteine residues and thereby prevents the formation of disulfide bonds due to non-specific oxidation during the lysis process (new Figure 3F). We also observed the increased dimer formation in old Pink1B9 flies under these conditions (new Figure 3F). Together these experiments indicate that the increased dimerization in old Pink1B9 flies is most probably caused by the formation of an intermolecular disulfide linking two Cisd molecules. As suggested by the reviewer, we have now added experiments that study the relationship between CISD1 dimerization and conventional mitophagy as described above (new Figure 2C).

There is a significant advancement. So far researchers were able to describe the importance of iron metabolism in PD (For example refer to work from the group of Georg Auburger such as PMID 33023155 and discussion of therapeutic intervention such as reviewed by Ma et al. PMID: 33799121) but few papers describe involvement of iron sulfur cluster proteins specifically (such as Aconitase) in relation to PINK1 and parkin (these are cited). The fact that CISD1 is a protein of the mitochondrial outer membrane makes it particularly interesting and further studies looking more closely at the interaction of CISD1 with mitochondrial proteins associated with PD will be of interest.

We thank the reviewer for pointing out these excellent publications. Key et al. present an enormous wealth of data on protein dysregulation of wildtype and Pink1-/- fibroblast cell lines upon perturbation of the iron homeostasis (Key *et al.,* 2020). Both cell lines exhibit a downregulation of CISD1 levels upon iron deprivation with the agent 2,2′ -Bipyridine possibly as a compensatory mechanism to limit the toxic gain of function of iron-depleted CISD1. The other paper, Ma et al. is a recent review on changes in iron homeostasis in PD and PD models (Ma *et al.,* 2021). Both papers are now cited in the manuscript.

Reviewer #2 (Evidence, reproducibility and clarity (Required)):Summary:In the paper entitled 'Mitochondrial CISD1 is a downstream target that mediates PINK1 and Parkin loss-of-function phenotypes', Bitar and co-workers investigate the interaction between CISD1 and the PINK1/Parkin pathway. Mutations in PINK1 and PARKIN cause early onset Parkinson's disease and CISD1 is a homodimeric mitochondrial iron-sulphur binding protein. They observed an increase in CISD1 dimer formation in dopaminergic neurons derived from Parkinson's disease patients carrying a PINK1 mutation. Immuno-blots of cells expressing CISD1 mutants that affects the iron sulphur cluster binding and as well as cells treated with iron chelators, showed that the tendency of CISD1 to form dimers is dependent on its binding to iron-sulphur clusters. Moreover, the Iron-depleted apo-CISD1 does not rescue mitochondrial phenotypes observed in CISD1 KO mouse cells. Finally, in vivo studies showed that overexpression of Cisd and mutant apo-Cisd in *Drosophila* shortened fly life span and, using a different overexpression model, apo-Cisd caused a delay in eclosion. Similar as patient derived neurons, they observed an increase in Cisd dimer levels in Pink1 mutant flies. Additionally, the authors showed that double mutants of Cisd and Pink1 alleviated all Pink1 mutant phenotypes, while double mutants of Prkn and Cisd rescued most Prkn mutant phenotypes.Major comments:The authors observed an increase in the levels of Cisd dimers in Pink1 mutant flies and removing Cisd in Pink1 mutant background rescues all the mutant phenotypes observed in Pink1 mutant flies, suggesting that the Cisd dimers are part and partial of the Pink1 mutant phenotype. The authors also generated a UAS_C111S_Cisd fly which can overexpress apo-Cisd. Overexpression of the C111S_Cisd construct with Tub-Gal4 showed a developmental delay. Since apo-Cisd forms more dimeric Cisd, my question is: does the strong overexpression (e.g. with Tub-Gal4) of the C111S_Cisd in wild type flies shows any of the Pink1 mutant phenotypes? If not, the authors should mention this and elaborate on it.

We thank the reviewer for their comments. We have exchanged these data with completely new data where we compared the effects of mild ubiquitous overexpression of mitochondrially targeted GFP with wildtype and mutant Cisd in wildtype and Pink1 mutant flies. We chose negative geotaxis (the ability to climb) and lifespan – known hallmarks of the *Pink1^B9^* phenotype – as readouts and found that *Pink1^B9^* flies have reduced negative geotaxis and lifespan as expected. Ubiquitous overexpression of Cisd and apo-Cisd aggravated these phenotypes in wildtype and Pink1 mutant flies (new Figure 4A). *Cisd C111S* overexpression was, however, associated with a more profound reduction in climbing speed compared to wildtype Cisd, while the effects on lifespan were similarly detrimental (new Figure 4A). Additionally, the vast majority of all *Cisd C111S*-overexpressing flies were not capable of inflating their wings (new Figure 4B), a rather non-specific phenotype which can result from genetic mutations, environmental conditions, and developmental defects. This phenotype did not occur in wildtype *Cisd* or *mitoGFP*-overexpressing flies (new Figure 4B).

Figure 6g: Shows the TEM pictures of the indirect flight muscles of Pink1 mutant flies and Pink1, Cisd double mutants. To me, the Picture of Pink1 mutant mitochondria is not very convincing. We expect swollen (enlarged) mitochondria with disrupted mitochondrial matrix. However, this is not clear in the picture. Moreover, in my opinion, Figure6 g, is missing an EM Picture of the Cisd mutant indirect flight muscles.

We now show exemplary pictures from Pink1 mutant and DKO in a higher magnification which better demonstrate the rounded Pink1 mutant mitochondria and the disrupted cristae structure. EM pictures of all four genotypes in different magnifications are now shown in new Figure 5 —figure supplement 4.

OPTIONAL: The authors suggest that most probably apo-Cisd, assumes a toxic function in Pink1 mutant flies and serves as a critical mediator of Pink1-linked phenotypes. If this statement is correct, we can hypothesize that increasing apo-Cisd in Pink1 mutant background should worsen the pink1 mutant defects.Therefore, I suggest overexpressing Cisd1 wild type (and/or C111S Cisd) in pink1 mutant flies, as pink1 is on the X chromosome, and mild overexpression of Cisd1 with da is not lethal, these experiments could be done in 3-4 fly crosses and hence within 1.5 – 2 months.

We have now done this experiment as mentioned above (new Figure 4).

Since Pink1 mutant flies contain higher levels of endogenous Cisd dimers, we can expect that overexpression of wild type Cisd will result in an even stronger increase of dimers. If these dimers indeed contribute to Pink1 mutant phenotypes we can expect that overexpression of Cisd will result in a worsening of the Pink1 mutant phenotypes.

We have now done this experiment as mentioned above and observed what the reviewer suspected (new Figure 4).

Minor Comments:In the Introduction (Background) there are some parts without references:E.g., there is not a single reference in the following part between'However, in unfit mitochondria with a reduced mitochondrial membrane potential …&… compromised mitochondria safeguards overall mitochondrial health and function.'

We thank the reviewer for pointing out this flaw. We have now added a suitable reference to the introduction.

In the introduction there is some confusion about the nomenclature used in the article: e.g. following comments are made in the text: Cisd2 (in this publication referred to as Dosmit) or fly Cisd2 (in this publication named MitoNEET).However, the names Dosmit and MitoNEET do not appear in the manuscript (except in references)

The literature and nomenclature for CISD1 are indeed confusing. We have now revised the introduction.

Figure 1: I am not sure why some gels are shown in this figure. The two last lanes of figure 1c are redundant and Figure 1c' which is also not mentioned in the text, is also a repetition of figure 1c.

The blots in 1c and 1c’ represent all data points (different patients and different individual differentiations) shown in the quantification in 1d. This is now explained better in the revised manuscript.

The authors mention in material and methods that T2A sites are used at the C-terminus of CISD1 to avoid tagging of CISD1. However, this is not entirely true as T2A will leave some amino acids (around 20) after the self-cleaving and therefore CISD1 will be tagged.

This is indeed true and we have now changed the wording in the revised manuscript.

In figure 5 P1 is used to abbreviate Pink1 mutants, however P1, to me, refers to pink1 wild type. It would be clearer to abbreviate Pink1 mutants as P1B9 in the graphs as B9 is the name of the mutant pink1 allele.

We thank the reviewer for pointing out this flaw. We have now altered Figure 5 to be clearer.

In figure 7: Parkin is abbreviated both as Prkn and as Park

We thank the reviewer for pointing out this flaw, we indeed mixed up both names because it is complicated. The gene symbol is Prkn, the fly line is called Park^25^. We have now clarified this in the text and new Figure 6.

I suggest changing the title. Recently an article (Ham et al., 2023 PMID: 37626046) was published showing similar genetic interactions between Pink1/Prkn and Cisd. However, the article of Ham et al., 2023 was focused on Pink1/Prkn regulation of ER calcium release, while this article is more related to iron homeostasis. I suggest that the title shows this distinction.

This is indeed a very good suggestion. We have now altered the title to “Iron/sulfur cluster loss of mitochondrial CISD1 mediates PINK1 loss-of-function phenotypes”.

Reviewer #2 (Significance (Required)):In general, this is an exciting and well-executed piece of research focusing on the defects in iron homeostasis observed in Parkinson's disease which a wide audience will appreciate. Very recently, a similar genetic interaction between Cisd and Pink1/Prkn in flies was published (Ham et al., 2023 PMID: 37626046) however, from a different angle. While, Ham et al. focused on the role of Pink1/Prkn and Cisd in IP3R related ER calcium release, this manuscript approaches the Pink1/Prkn – Cisd interaction from an iron homeostasis point of view. Since, iron dysregulation contributes to the pathogenesis of Parkinson's disease, the observations in this manuscript are relevant for the disease. Hence, the work is sufficiently novel and deserves publication. However, additional experiments are suggested to strengthen the authors' conclusions.

We thank the reviewer for their kind words. As mentioned above, these additional experiments are now included in our revised manuscript (v2).

I work on *Drosophila* models of Parkinson's diseaseReferees cross-commentingI agree with the reviewer number 1 that it would be interesting to investigate CISD1 dimerisation status during mitophagy.

As mentioned above, we now studied CISD1 dimerization in response to more classical activators of mitophagy namely FCCP and antimycin/oligomycin which had no significant effect on dimerization suggesting that this phenotype is more pronounced under iron depletion. These data are shown in the new Figure 2C.

Reviewer #3 (Evidence, reproducibility and clarity (Required)):Here the authors provide evidence that Cisd is downstream of Parkin/Pink1 and suggest that the levels of apo-Cisd correlate with neurotoxicity. The data presented generally supports the conclusions of the authors and will be useful to those in the field. The manuscript would be improved by a more balanced discussion of the strengths and weaknesses of the study and more circumspection in interpretation of data.

We thank the reviewer for their comments aimed to improve our manuscript. We have now discussed the strengths and weaknesses of our study in more detail.

Introduction. While iron has been implicated in Parkinson's disease, it is an overstatement to say that disruption in iron metabolism contributes significantly to the pathogenesis of the disease.

There is certainly a plethora of data implicating perturbed iron homeostasis in PD as also pointed out by reviewer #1. We have tried to tone down our wording in the text and added a recent review on the topic (Ma *et al.,* 2021) as also suggested by reviewer #1.

Introduction. The discussion of the various names for Cisd2 is important, but confusing as written. Specifically, the use of "this" makes the wording unclear.

We thank the reviewer for pointing out this flaw. We have altered the wording in the introduction.

Methods. It would be preferable to use heterozygous driver lines or a more similar genetic control rather than w-1118.

We have exchanged these data with completely new data where we compared the effects of mild ubiquitous overexpression of mitochondrially targeted GFP with wildtype and mutant Cisd in wildtype and Pink1 mutant flies. We chose negative geotaxis (the ability to climb) and lifespan – known hallmarks of the *Pink1^B9^* phenotype – as readouts and found that *Pink1^B9^* flies have reduced negative geotaxis and lifespan as expected. Ubiquitous overexpression of Cisd and apo-Cisd aggravated these phenotypes in wildtype and Pink1 mutant flies (new Figure 4A). *Cisd C111S* overexpression was, however, associated with a more profound reduction in climbing speed compared to wildtype Cisd, while the effects on lifespan were similarly detrimental (new Figure 4A). Additionally, the vast majority of all *Cisd C111S*-overexpressing flies were not capable of inflating their wings (new Figure 4B), a rather non-specific phenotype which can result from genetic mutations, environmental conditions, and developmental defects. This phenotype did not occur in wildtype *Cisd* or *mitoGFP*-overexpressing flies (new Figure 4B).

The data shown in Figure 5, 6, and 7 all used *w^1118^* as control because all other fly strains are on the same genetic background.

Page 10. It appears that the PINK1 lines have been described previously. The authors should clarify this point and ensure that the new data presented in the current manuscript (presumably the mRNA levels, Figure 1a) is indicated, as well as data that is confirmatory of prior findings (Figure 1b).

Yes, these PINK1 lines have been described previously as pointed out in the manuscript. The original paper did not quantify the PINK1 mRNA levels shown in Figure 1a. The blots shown in Figure 1b are from new differentiations and have also not been shown before but confirm findings published in Jarazo et al. (Jarazo *et al.,* 2022). This has been clarified in the revised version of our manuscript.

Figure 3 legend. There is a typographical error, "ne-way ANOVA."

We thank the reviewer for pointing out this flaw. This has been corrected in the revised version.

Page 15. The nature of the Pink1-B9 mutant should be specified.

We now added a figure 3 —figure supplement 1 that depicts the specific mutation in these flies.

Figure 4. Levels of mutant and wild type Cisd should be compared in transgenic flies.

Levels of Cisd expressed in da>>Cisd C111S are lower than in flies overexpressing wildtype Cisd or mitoGFP which probably reflects enhanced degradation. Because we now used mitoGFP as a control, we did not follow up on this.

Figure 5b,d. The striking change seems to be the decrease in dimers in young Pink1 mutant animals, not the small increase in dimers in the older Pink1 mutants.

It is always difficult to find a “typical” picture that reflects all changes observed in quantitative data. This Figure actually shows a decrease of total Cisd levels in young flies in Figure 5c but no difference of the dimer/monomer ratio in Figure 5d. These data are now shown in Figure 3. We also added new supporting data in new Figure 3E and F.

Figure 5f. Caution should be used in interpreting the results. Deferiprone has toxicity to wildtype flies (trend) and may simply be making sick Pink1 mutants sicker.

There is certainly a tendency for wildtype flies to thrive less in food containing deferiprone. To make this more obvious, we have now added the exact *p* value (0.0764, which we don’t consider borderline but a tendency) to this figure and mention this fact in the text.

Figure 5e. The data are hard to interpret. The number of animals is very small for a viability study and the strains are apparently in different genetic backgrounds, though this is not clearly specified. The experiment in Supplementary Figure 1 appears better controlled and supports the Pink1 data; however, a similar concern pertains to Figure 7. The authors may thus wish to be more circumspect in their interpretation, especially of the Parkin data.

In Figure 5e we quantified total iron levels and the Fe^3+^/Fe^2+^ ratio using capillary electrophoresis-inductively coupled plasma mass spectrometry (CE-ICP-MS). Although indeed not so many flies were used in this quantification, the results are highly significant. If the reviewer was referring to Figure 5f, we agree that this experiment was not well (to be honest, even wrongly explained) which we corrected in the revised version of this manuscript. We thank the reviewer for pointing out this flaw.

Reviewer #3 (Significance (Required)):The major significance of the study is in putting downstream of Parkin/Pink1 (largely confirmatory to recently published work) and suggesting that the levels of apo-Cisd are an important determinant of toxicity. The work will be of interest to those in the field.Description of analyses that authors prefer not to carry out

All changes suggested by the reviewers were addressed (v1) or will be addressed (v2).

References

Jarazo J, Barmpa K, Modamio J, Saraiva C, Sabaté-Soler S, Rosety I, Griesbeck A, Skwirblies F, Zaffaroni G, Smits LM, *et al.* (2022) Parkinson’s Disease Phenotypes in Patient Neuronal Cultures and Brain Organoids Improved by 2-Hydroxypropyl-β-Cyclodextrin Treatment. *Mov Disord* 37: 80–94

Key J, Sen NE, Arsović A, Krämer S, Hülse R, Khan NN, Meierhofer D, Gispert S, Koepf G & Auburger G (2020) Systematic Surveys of Iron Homeostasis Mechanisms Reveal Ferritin Superfamily and Nucleotide Surveillance Regulation to be Modified by PINK1 Absence. *Cells* 9

Ma L, Gholam Azad M, Dharmasivam M, Richardson V, Quinn RJ, Feng Y, Pountney DL, Tonissen KF, Mellick GD, Yanatori I, *et al.* (2021) Parkinson’s disease: Alterations in iron and redox biology as a key to unlock therapeutic strategies. *Redox Biol* 41: 101896